# YELLOWFIN AND THE ART OF MOMENTUM TUNING

## ABSTRACT

Hyperparameter tuning is one of the most time-consuming workloads in deep learning. State-of-the-art optimizers, such as AdaGrad, RMSProp and Adam, reduce this labor by adaptively tuning an individual learning rate for each variable. Recently researchers have shown renewed interest in simpler methods like momentum SGD as they may yield better results. Motivated by this trend, we ask: can simple adaptive methods, based on SGD perform as well or better? We revisit the momentum SGD algorithm and show that hand-tuning a single learning rate and momentum makes it competitive with Adam. We then analyze its robustness to learning rate misspecification and objective curvature variation. Based on these insights, we design YELLOWFIN, an automatic tuner for momentum and learning rate in SGD. YELLOWFIN optionally uses a negative-feedback loop to compensate for the momentum dynamics in asynchronous settings on the fly. We empirically show YELLOWFIN can converge in fewer iterations than Adam on ResNets and LSTMs for image recognition, language modeling and constituency parsing, with a speedup of up to 3.28x in synchronous and up to 2.69x in asynchronous settings.

## 1 INTRODUCTION

Accelerated forms of stochastic gradient descent (SGD), pioneered by Polyak (1964) and Nesterov (1983), are the de-facto training algorithms for deep learning. Their use requires a sane choice for their *hyperparameters*: typically a *learning rate* and *momentum parameter* (Sutskever et al., 2013). However, tuning hyperparameters is arguably the most time-consuming part of deep learning, with many papers outlining best tuning practices written (Bengio, 2012; Orr and Müller, 2003; Bengio et al., 2012; Bottou, 2012). Deep learning researchers have proposed a number of methods to deal with hyperparameter optimization, ranging from grid-search and smart black-box methods (Bergstra and Bengio, 2012; Snoek et al., 2012) to adaptive optimizers. Adaptive optimizers aim to eliminate hyperparameter search by tuning on the fly for a single training run: algorithms like AdaGrad (Duchi et al., 2011), RMSProp (Tieleman and Hinton, 2012) and Adam (Kingma and Ba, 2014) use the magnitude of gradient elements to tune learning rates *individually for each variable* and have been largely successful in relieving practitioners of tuning the learning rate.

Recently some researchers have started favoring simple momentum SGD over the previously mentioned adaptive methods (Chen et al., 2016; Gehring et al., 2017), often reporting better test scores (Wilson et al., 2017). Motivated by this trend, we ask the question: can simpler adaptive methods, based on momentum SGD perform as well or better? We empirically show that, with hand-tuned learning rate, Polyak's momentum SGD achieves faster convergence than Adam for a large class of models. We then formulate the optimization update as a dynamical system and study certain robustness properties

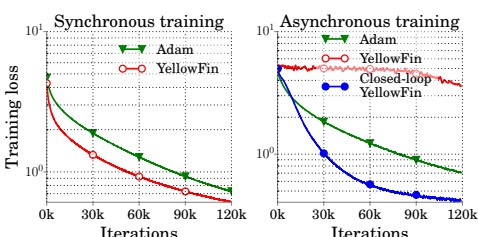

Figure 1: YELLOWFIN in comparison to Adam on a ResNet (CIFAR100, cf. Section 5).

of the momentum operator. Building on our analysis, we design YELLOWFIN, an automatic hyperparameter tuner for momentum SGD. YELLOWFIN simultaneously tunes the learning rate and momentum on the fly, and can handle the complex dynamics of asynchronous execution. Specifically:

- In Section 2, we show that momentum presents convergence robust to learning rate misspecification and curvature variation in a class of non-convex objectives; this robustness is desirable for

deep learning. They stem from a known but obscure fact: the momentum operator's spectral radius is constant in a large subset of the hyperparameter space.

- In Section 3, we use these robustness insights and a simple quadratic model analysis to design YELLOWFIN, an automatic tuner for momentum SGD. YELLOWFIN uses on-the-fly measurements from the gradients to tune both a single learning rate and momentum.

- In Section 3.3, we discuss common stability concerns related to the phenomenon of exploding gradients (Pascanu et al., 2013). We present a natural extension to our basic tuner, using adaptive gradient clipping, to stabilize training for objectives with exploding gradients.

- In Section 4 we present closed-loop YELLOWFIN, suited for asynchronous training. It uses a novel component for measuring the total momentum in a running system, including any asynchrony-induced momentum, a phenomenon described in Mitliagkas et al. (2016). This measurement is used in a negative feedback loop to control the value of algorithmic momentum.

We provide a thorough evaluation of the performance and stability of our tuner. In Section 5, we demonstrate empirically that on ResNets and LSTMs YELLOWFIN can converge in fewer iterations compared to: (i) hand-tuned momentum SGD (up to $1.75$x speedup); and (ii) default Adam ($0.8$x to $3.3$x speedup). Under asynchrony, the closed-loop control architecture speeds up YELLOWFIN, making it up to $2.69$x faster than Adam. Our experiments include runs on 7 different models, randomized over at least 5 different random seeds. YELLOWFIN is stable and achieves consistent performance: the normalized sample standard deviation of test metrics varies from $0.05\%$ to $0.6\%$. We released PyTorch and TensorFlow implementations, that can be used as drop-in replacements for any optimizer. YELLOWFIN has also been implemented in other various packages. Its large-scale deployment in industry has taught us important lessons about stability; we discuss those challenges and our solution in Section 3.3. We conclude with related work and discussion in Section 6 and 7.

## 2 THE MOMENTUM OPERATOR

In this section we identify the main technical insights guiding the design of YELLOWFIN. We show that momentum presents convergence robust to learning rate misspecification and curvature variation for a class of non-convex objectives; these robustness properties are desirable for deep learning.

### 2.1 PRELIMINARIES

We aim to minimize some objective $f(x)$. In machine learning, $x$ is referred to as *the model* and the objective is some *loss function*. A low loss implies a well-fit model. Gradient descent-based procedures use the gradient of the objective function, $\nabla f(x)$, to update the model iteratively. Polyak's momentum gradient descent update (Polyak, 1964) is given by

$$x_{t+1} = x_t - \alpha \nabla f(x_t) + \mu(x_t - x_{t-1}), \tag{1}$$

where $\alpha$ denotes the learning rate and $\mu$ the value of momentum used. Momentum's main appeal is its established ability to *accelerate convergence* (Polyak, 1964). On a strongly convex smooth function with condition number $\kappa$, the optimal convergence rate of gradient descent ($\mu = 0$) is $O(\frac{\kappa-1}{\kappa+1})$ (Nesterov, 2013). On the other hand, for certain classes of strongly convex and smooth functions, like quadratics, the optimal momentum value,

$$\mu^* = \left(\frac{\sqrt{\kappa}-1}{\sqrt{\kappa}+1}\right)^2, \tag{2}$$

yields the optimal accelerated rate $O(\frac{\sqrt{\kappa}-1}{\sqrt{\kappa}+1})$. [1] This is the smallest value of momentum that **ensures the same rate of convergence along all directions**. This fact is often hidden away in proofs. We shed light on some of its previously unknown implications in Section 2.2.

### 2.2 ROBUSTNESS PROPERTIES OF THE MOMENTUM OPERATOR

In this section we analyze the dynamics of momentum on a simple class of one dimensional, non-convex objectives. We first introduce the notion of *generalized curvature* and use it to describe the momentum operator. Then we discuss some properties of the momentum operator.

---

[1]This guarantee does not generalize to arbitrary strongly convex functions (Lessard et al., 2016). Nonetheless, acceleration is often observed in practice (cf. Section 2.2).

**Definition 1** (Generalized curvature). *The derivative of $f(x) : \mathbb{R} \to \mathbb{R}$, can be written as*

$$f'(x) = h(x)(x - x^*) \tag{3}$$

*for some $h(x) \in \mathbb{R}$, where $x^*$ is the global minimum of $f(x)$. We call $h(x)$ the* generalized curvature.

The generalized curvature describes, in some sense, curvature with respect to the optimum, $x^*$. For quadratic objectives, it coincides with the standard definition of curvature, and is the sole quantity related to the objective that influences the dynamics of gradient descent. For example, the contraction of a gradient descent step is $1 - \alpha h(x_t)$. Let $\boldsymbol{A}_t$ denote the *momentum operator* at time $t$. Using a state-space augmentation, we can express the momentum update as

$$\begin{pmatrix} x_{t+1} - x^* \\ x_t - x^* \end{pmatrix} = \begin{bmatrix} 1 - \alpha h(x_t) + \mu & -\mu \\ 1 & 0 \end{bmatrix} \begin{pmatrix} x_t - x^* \\ x_{t-1} - x^* \end{pmatrix} \triangleq \boldsymbol{A}_t \begin{pmatrix} x_t - x^* \\ x_{t-1} - x^* \end{pmatrix}. \tag{4}$$

**Lemma 2** (Robustness of the momentum operator). *As proven in Appendix A, if the generalized curvature, $h$, and hyperparameters $\alpha, \mu$ are in the* robust region*, that is:*

$$(1 - \sqrt{\mu})^2 \leq \alpha h(x_t) \leq (1 + \sqrt{\mu})^2, \tag{5}$$

*then the spectral radius of the momentum operator only depends on momentum: $\rho(\boldsymbol{A}_t) = \sqrt{\mu}$.*

We explain Lemma 2 as robust convergence with respect to learning rate and to variations in curvature.

**Momentum is robust to learning rate misspecification** For a one dimensional strongly convex quadratic objective, we get $h(x) = h$ for all $x$ and Lemma 2 suggests that $\rho(\boldsymbol{A}_t) = \sqrt{\mu}$ as long as

$$(1 - \sqrt{\mu})^2 / h \leq \alpha \leq (1 + \sqrt{\mu})^2 / h. \tag{6}$$

In Figure 2, we plot $\rho(\boldsymbol{A}_t)$ for different $\alpha$ and $\mu$. As we increase the value of momentum, the optimal rate of convergence $\sqrt{\mu}$ is achieved by an ever-widening range of learning rates. Furthermore, for objectives with large condition number, higher values of momentum are *both faster and more robust*. **This property influences the design of our tuner:** as long as the learning rate satisfies (6), we are in the *robust region* and expect the same asymptotics, e.g. a convergence rate of $\sqrt{\mu}$ for quadratics, independent of the learning rate. Having established that, we can just focus on optimally tuning momentum.

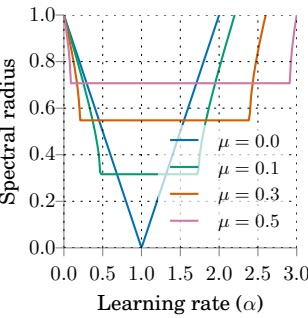

Figure 2: Momentum operator on scalar quadratic.

**Momentum is robust to curvature variation** As discussed in Section 2.1, the intuition hidden in classic results is that for strongly convex smooth objectives, the momentum value in (2) guarantees the same rate of convergence along all directions. We extend this intuition to certain non-convex functions. Lemma 2 guarantees a constant, time-homogeneous spectral radius for the momentum operators $(\boldsymbol{A}_t)_t$ if (5) is satisfied at every step. This motivates an extension of the condition number.

**Definition 3** (Generalized condition number). *We define the generalized condition number (GCN) of a scalar function, $f(x) : \mathbb{R} \to \mathbb{R}$, to be the dynamic range of its generalized curvature, $h(x)$:*

$$\nu = \frac{\sup_{x \in dom(f)} h(x)}{\inf_{x \in dom(f)} h(x)} \tag{7}$$

The GCN captures variations in generalized curvature along a scalar slice. From Lemma 2 we get

$$\mu^* = \left( \frac{\sqrt{\nu} - 1}{\sqrt{\nu} + 1} \right)^2, \quad \alpha^* = \frac{(1 - \sqrt{\mu})^2}{\inf_{x \in dom(f)} h(x)} \tag{8}$$

as the optimal hyperparameters. Specifically, $\mu^*$ is the smallest momentum value that guarantees a homogeneous spectral radius of $\sqrt{\mu^*}$ for all $(\boldsymbol{A}_t)_t$. The spectral radius of an operator describes its asymptotic convergence behavior. However, the product of a sequence of operators $\boldsymbol{A}_t \cdots \boldsymbol{A}_1$ all with spectral radius $\sqrt{\mu}$ does not always follow the asymptotics of $\sqrt{\mu}^t$. In other words, *we do not provide a convergence rate guarantee*. Instead, we provide evidence in support of this intuition. For example, the non-convex objective in Figure 3(a), composed of two quadratics with curvatures 1 and 1000, has a GCN of 1000. Using the tuning rule of (8), and running the momentum algorithm (Figure 3(b)) yields a practically constant rate of convergence throughout. In Figures 3(c,d) we demonstrate that for an LSTM, the majority of variables follow a $\sqrt{\mu}$ convergence rate. **This property influences the design of our tuner:** in the next section we use the tuning rules of (8) in YELLOWFIN, generalized appropriately to handle SGD noise.

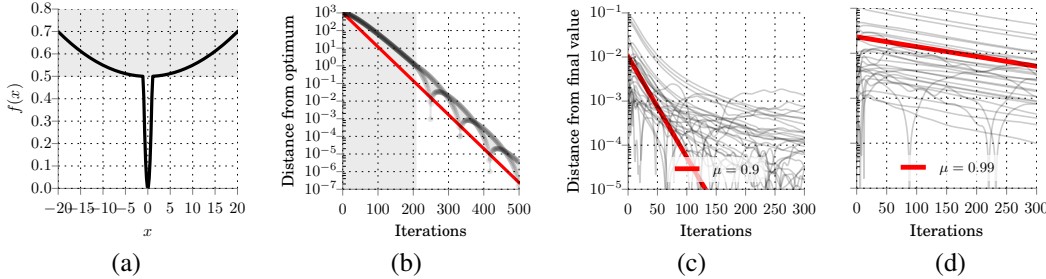

Figure 3: (a) Non-convex toy example; (b) constant convergence rate achieved empirically on the objective of (a) tuned according to (8); (c,d) LSTM on MNIST: as momentum increases, more variables (shown in grey) fall in the robust region and follow the robust rate, $\sqrt{\mu}$.

## 3 THE YELLOWFIN TUNER

In this section we describe YELLOWFIN, our tuner for momentum SGD. We introduce a noisy quadratic model and work on a local quadratic approximation of $f(x)$ to apply the tuning rule of (8) to SGD on an arbitrary objective. YELLOWFIN is our implementation of that rule.

**Noisy quadratic model**  We consider minimizing the one-dimensional quadratic

$$f(x) = \frac{1}{2}hx^2 + C = \frac{1}{n}\sum_i \frac{1}{2}h(x - c_i)^2 \triangleq \frac{1}{n}\sum_i f_i(x), \quad \sum_i c_i = 0. \quad (9)$$

The objective is defined as the average of $n$ *component functions*, $f_i$. This is a common model for SGD, where we use only a single data point (or a mini-batch) drawn uniformly at random, $S_t \sim \text{Uni}([n])$ to compute a noisy gradient, $\nabla f_{S_t}(x)$, for step $t$. Here, $C = \frac{1}{2n}\sum_i hc_i^2$ denotes the *gradient variance*. This scalar model is sufficient to study an arbitrary local quadratic approximation: optimization on quadratics decomposes trivially into scalar problems along the principal eigenvectors of the Hessian. Next we get an *exact* expression for the mean square error after running momentum SGD on a scalar quadratic for $t$ steps.

**Lemma 4.** *Let $f(x)$ be defined as in* (9), *$x_1 = x_0$ and $x_t$ follow the momentum update* (1) *with stochastic gradients $\nabla f_{S_t}(x_{t-1})$ for $t \geq 2$. Let $e_1 = [1, 0]^T$, the expectation of squared distance to the optimum $x^*$ is*

$$\mathbb{E}(x_{t+1} - x^*)^2 = (e_1^\top A^t [x_1 - x^*, x_0 - x^*]^\top)^2 + \alpha^2 C e_1^\top (I - B^t)(I - B)^{-1} e_1, \quad (10)$$

*where the first and second term correspond to squared bias and variance, and their corresponding momentum dynamics are captured by operators*

$$A = \begin{bmatrix} 1 - \alpha h + \mu & -\mu \\ 1 & 0 \end{bmatrix}, \quad B = \begin{bmatrix} (1 - \alpha h + \mu)^2 & \mu^2 & -2\mu(1 - \alpha h + \mu) \\ 1 & 0 & 0 \\ 1 - \alpha h + \mu & 0 & -\mu \end{bmatrix}. \quad (11)$$

Even though it is possible to numerically work on (10) directly, we use a scalar, asymptotic surrogate based in (12) on the spectral radii of operators to simplify analysis and expose insights. This decision is supported by our findings in Section 2: the spectral radii can capture empirical convergence speed.

$$\mathbb{E}(x_{t+1} - x^*)^2 \approx \rho(A)^{2t}(x_0 - x_*)^2 + (1 - \rho(B)^t)\frac{\alpha^2 C}{1 - \rho(B)} \quad (12)$$

One of our design decisions for YELLOWFIN is to always work in the robust region of Lemma 2. We know that this implies a spectral radius $\sqrt{\mu}$ of the momentum operator, $A$, for the bias. Lemma 5 shows that under the exact same condition, the variance operator $B$ has spectral radius $\mu$.

**Lemma 5.** *The spectral radius of the variance operator, $B$ is $\mu$, if $(1 - \sqrt{\mu})^2 \leq \alpha h \leq (1 + \sqrt{\mu})^2$.*

As a result, the surrogate objective of (12), takes the following form in the robust region.

$$\mathbb{E}(x_{t+1} - x^*)^2 \approx \mu^t(x_0 - x^*)^2 + (1 - \mu^t)\frac{\alpha^2 C}{1 - \mu} \quad (13)$$

We use this surrogate objective to extract a noisy tuning rule for YELLOWFIN.

### 3.1 TUNING RULE

Based on the surrogate in (13), we present YELLOWFIN (Algorithm 1). Let $D$ denote an estimate of the current model's distance to a local quadratic approximation's minimum, and $C$ denote an estimate for gradient variance. Also, let $h_{max}$ and $h_{min}$ denote estimates for the largest and smallest generalized curvature respectively. The extremal curvatures $h_{min}$ and $h_{max}$ are meant to capture both curvature variation along different directions (like the classic condition number) and also variation that occurs as the *landscape evolves*. At each iteration, we solve the following SINGLESTEP problem.

(SINGLESTEP)

$$\mu_t, \alpha_t = \arg\min_{\mu} \mu D^2 + \alpha^2 C$$

$$s.t. \quad \mu \geq \left( \frac{\sqrt{h_{\max}/h_{\min}} - 1}{\sqrt{h_{\max}/h_{\min}} + 1} \right)^2$$

$$\alpha = \frac{(1 - \sqrt{\mu})^2}{h_{\min}}$$

---
**Algorithm 1** YELLOWFIN

**function** YELLOWFIN(gradient $g_t$, $\beta$)
    $h_{\max}, h_{\min} \leftarrow$ CURVATURERANGE($g_t, \beta$)
    $C \leftarrow$ VARIANCE($g_t, \beta$)
    $D \leftarrow$ DISTANCE($g_t, \beta$)
    $\mu_t, \alpha_t \leftarrow$ SINGLESTEP($C, D, h_{\max}, h_{\min}$)
    **return** $\mu_t, \alpha_t$
**end function**

---

SINGLESTEP minimizes the surrogate for the expected squared distance from the optimum of a local quadratic approximation (13) after a single step ($t = 1$), while keeping all directions in the robust region (5). This is the SGD version of the noiseless tuning rule in (8). It can be solved in closed form; we refer to Appendix D for discussion on the closed form solution. YELLOWFIN uses functions CURVATURERANGE, VARIANCE and DISTANCE to measure quantities $h_{\max}$, $h_{\min}$, $C$ and $D$ respectively. These measurement functions can be designed in different ways. We present the implementations we used for our experiments, based completely on gradients, in Section 3.2.

### 3.2 MEASUREMENT FUNCTIONS IN YELLOWFIN

This section describes our implementation of the measurement oracles used by YELLOWFIN: CURVATURERANGE, VARIANCE, and DISTANCE. We design the measurement functions with the assumption of a negative log-probability objective; this is in line with typical losses in machine learning, e.g. cross-entropy for neural nets and maximum likelihood estimation in general. Under this assumption, the Fisher information matrix—i.e. the expected outer product of noisy gradients—equals the Hessian of the objective (Duchi, 2016; Pascanu and Bengio, 2013). This allows for measurements purely from minibatch gradients with overhead linear to model dimensionality. These implementations are not guaranteed to give accurate measurements. Nonetheless, their use in our experiments in Section 5 shows that they are sufficient for YELLOWFIN to outperform the state of the art on a variety of objectives. We also refer to Appendix E for implementation details on zero-debias (Kingma and Ba, 2014), slow start (Schaul et al., 2013) and smoothing for curvature range estimation.

---
**Algorithm 2** Curvature range

**state:** $h_{\max}, h_{\min}, h_i, \forall i \in \{1, 2, 3, ...\}$
**function** CURVATURERANGE(gradient $g_t, \beta$)
  $h_t \leftarrow \|g_t\|^2$
  $h_{\max,t} \leftarrow \max_{t-w \leq i \leq t} h_i$, $h_{\min,t} \leftarrow \min_{t-w \leq i \leq t} h_i$
  $h_{\max} \leftarrow \beta \cdot h_{\max} + (1 - \beta) \cdot h_{\max,t}$
  $h_{\min} \leftarrow \beta \cdot h_{\min} + (1 - \beta) \cdot h_{\min,t}$
  **return** $h_{\max}, h_{\min}$
**end function**

**Algorithm 3** Gradient variance

**state:** $\overline{g^2} \leftarrow 0, \overline{g} \leftarrow 0$
**function** VARIANCE(gradient $g_t, \beta$)
  $\overline{g^2} \leftarrow \beta \cdot \overline{g^2} + (1 - \beta) \cdot g_t \odot g_t$
  $\overline{g} \leftarrow \beta \cdot \overline{g} + (1 - \beta) \cdot g_t$
  **return** $\|\overline{g^2} - \overline{g}^2\|_1$
**end function**

**Algorithm 4** Distance to opt.

**state:** $\overline{\|g\|} \leftarrow 0, \overline{h} \leftarrow 0$
**function** DISTANCE(gradient $g_t, \beta$)
  $\overline{\|g\|} \leftarrow \beta \cdot \overline{\|g\|} + (1 - \beta) \cdot \|g_t\|$
  $\overline{h} \leftarrow \beta \cdot \overline{h} + (1 - \beta) \cdot \|g_t\|^2$
  $D \leftarrow \beta \cdot D + (1 - \beta) \cdot \overline{\|g\|}/\overline{h}$
  **return** $D$
**end function**

---

**Curvature range** Let $g_t$ be a noisy gradient, we estimate the range of curvatures in Algorithm 2. We notice that the outer product of $g_t$ and $g_t^T$ has an eigenvalue $h_t = \|g_t\|^2$ with eigenvector $g_t$. Thus under our negative log-likelihood assumption, we use $h_t$ to approximate the curvature of Hessian along gradient direction $g_t$. Specifically, we maintain $h_{\min}$ and $h_{\max}$ as running averages of extreme curvature $h_{\min,t}$ and $h_{\max,t}$, from a sliding window of width 20. As gradient directions evolve, we explore curvatures along different directions. Thus $h_{\min}$ and $h_{\max}$ capture the curvature variations.

**Gradient variance** To estimate the gradient variance in Algorithm 3, we use running averages $\overline{g}$ and $\overline{g^2}$ to keep track of $g_t$ and $g_t \odot g_t$, the first and second order moment of the gradient. As $\text{Var}(g_t) = \mathbb{E}g_t^2 - (\mathbb{E}g_t)^2$, we estimate the gradient variance $C$ in (14) using $C = \|\overline{g^2} - \overline{g}^2\|_1$.

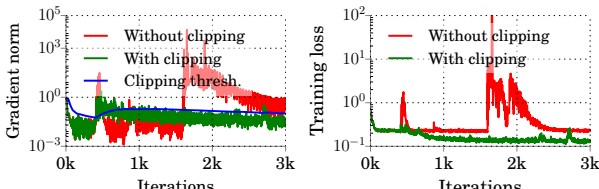

Figure 4: A variation of the LSTM architecture in (Zhu et al., 2016) exhibits exploding gradients. The proposed adaptive gradient clipping threshold (blue) stabilizes the training loss.

|  | Loss | BLEU4 |
|---|---|---|
| Default w/o clip. | diverge | |
| Default w/ clip. | 2.86 | 30.75 |
| YF | **2.75** | **31.59** |

Table 1: German-English translation validation performance using convolutional seq-to-seq learning.

**Distance to optimum** In Algorithm 4, we estimate the distance to the optimum of the local quadratic approximation. Inspired by the fact that $\|\nabla f(\boldsymbol{x})\| \leq \|\boldsymbol{H}\|\|\boldsymbol{x} - \boldsymbol{x}^\star\|$ for a quadratic $f(x)$ with Hessian $\boldsymbol{H}$ and minimizer $\boldsymbol{x}^*$, we first maintain $\overline{h}$ and $\overline{\|g\|}$ as running averages of curvature $h_t$ and gradient norm $\|g_t\|$. Then the distance is approximated using $\overline{\|g\|}/\overline{h}$.

### 3.3 Stability on non-smooth objectives

Neural network objectives can involve arbitrary non-linearities, and large Lipschitz constants (Szegedy et al., 2013). Furthermore, the process of training them is inherently non-stationary, with the landscape abruptly switching from flat to steep areas. In particular, the objective functions of RNNs with hidden units can exhibit occasional but very steep slopes (Pascanu et al., 2013). To deal with this issue, we use *adaptive gradient clipping* heuristics as a very natural addition to our basic tuner. It is discussed with extensive details in Appendix F. In Figure 4, we present an example of an LSTM that exhibits the 'exploding gradient' issue. The proposed adaptive clipping can stabilize the training process using YELLOWFIN and prevent large catastrophic loss spikes.

We validate the proposed adaptive clipping on the convolutional sequence to sequence learning model (Gehring et al., 2017) for IWSLT 2014 German-English translation. The default optimizer (Gehring et al., 2017) uses learning rate 0.25 and Nesterov's momentum 0.99, diverging to loss overflow due to 'exploding gradient'. It requires, as in Gehring et al. (2017), strict manually set gradient norm threshold 0.1 to stabilize. In Table 1, we can see YellowFin, with adaptive clipping, outperforms the default optimizer using manually set clipping, with 0.84 higher validation BLEU4 after 120 epochs.

## 4 Closed-loop YellowFin

Asynchrony is a parallelization technique that avoids synchronization barriers (Niu et al., 2011). It yields better hardware efficiency, i.e. faster steps, but can increase the number of iterations to a given metric, i.e. statistical efficiency, as a tradeoff (Zhang and Ré, 2014). Mitliagkas et al. (2016) interpret asynchrony as added momentum dynamics. We design closed-loop YELLOWFIN, a variant of YELLOWFIN to automatically control algorithmic momentum, compensate for asynchrony and accelerate convergence. We use the formula in (14) to model the dynamics in the system, where the total momentum, $\mu_T$, includes both asynchrony-induced and algorithmic momentum, $\mu$, in (1).

$$\mathbb{E}[x_{t+1} - x_t] = \mu_T \mathbb{E}[x_t - x_{t-1}] - \alpha \mathbb{E}\nabla f(x_t) \tag{14}$$

We first use (14) to design an robust estimator $\hat{\mu}_T$ for the value of total momentum at every iteration. Then we use a simple negative feedback control loop to adjust the value of algorithmic momentum so that $\hat{\mu}_T$ matches the *target momentum* decided by YELLOWFIN in Algorithm 1. In Figure 5, we demonstrate momentum dynamics in an asynchronous training system. As directly using the target value as algorithmic momentum, YELLOWFIN (middle) presents total momentum $\hat{\mu}_T$ strictly larger than the target momentum, due to asynchrony-induced momentum. Closed-loop YELLOWFIN (right) automatically brings down algorithmic momentum, match measured total momentum $\hat{\mu}_T$ to target value and, as we will see, significantly speeds up convergence comparing to YELLOWFIN. We refer to Appendix G for details on estimator $\hat{\mu}_T$ and Closed-loop YELLOWFIN in Algorithm 5.

## 5 Experiments

In this section, we empirically validate the importance of momentum tuning and evaluate YELLOWFIN in both synchronous (single-node) and asynchronous settings. In synchronous settings, we first demonstrate that, with hand-tuning, momentum SGD is competitive with Adam, a state-of-the-art

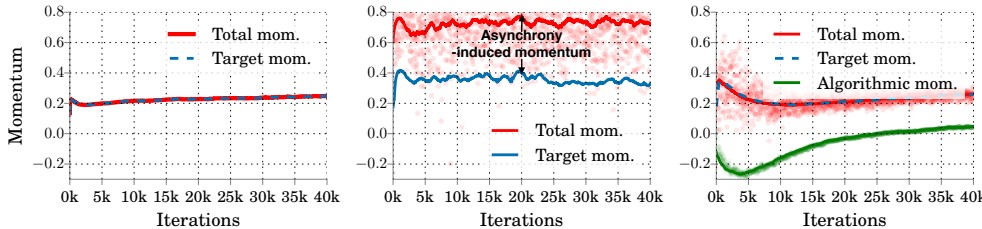

Figure 5: When running YELLOWFIN, total momentum $\hat{\mu}_t$ equals algorithmic value in synchronous settings (left); $\hat{\mu}_t$ is greater than algorithmic value on 16 asynchronous workers (middle). Closed-loop YELLOWFIN automatically lowers algorithmic momentum and brings total momentum to match the target value (right). Red dots are measured $\hat{\mu}_t$ at every step with red line as its running average.

adaptive method. Then, we evaluate YELLOWFIN *without any hand tuning* in comparison to hand-tuned Adam and momentum SGD. In asynchronous settings, we show that closed-loop YELLOWFIN accelerates with momentum closed-loop control, performing significantly better than Adam.

We evaluate on convolutional neural networks (CNN) and recurrent neural networks (RNN). For CNN, we train ResNet (He et al., 2016) for image recognition on CIFAR10 and CIFAR100 (Krizhevsky et al., 2014). For RNN, we train LSTMs for character-level language modeling with the TinyShakespeare (TS) dataset (Karpathy et al., 2015), word-level language modeling with the Penn TreeBank (PTB) (Marcus et al., 1993), and constituency parsing on the Wall Street Journal (WSJ) dataset (Choe and Charniak). We refer to Table 3 in Appendix H for model specifications. *To eliminate influences of a specific random seed, in our synchronous and asynchronous experiments, the training loss and validation metrics are averaged from 3 runs using different random seeds.*

## 5.1 SYNCHRONOUS EXPERIMENTS

We tune Adam and momentum SGD on learning rate grids with prescribed momentum 0.9 for SGD. We fix the parameters of Algorithm 1 in all experiments, i.e. YELLOWFIN runs *without any hand tuning*. We provide full specifications, including the learning rate (grid) and the number of iterations we train on each model in Appendix I. For visualization purposes, we smooth training losses with a uniform window of width 1000. For Adam and momentum SGD on each model, we pick the configuration achieving the lowest averaged smoothed loss. To compare two algorithms, we record the lowest smoothed loss achieved by both. Then the speedup is reported as the ratio of iterations to achieve this loss. We use this setup to validate our claims.

**Momentum SGD is competitive with adaptive methods** In Table 2, we compare tuned momentum SGD and tuned Adam on ResNets with training losses shown in Figure 9 in Appendix J. We can observe that momentum SGD achieves 1.71x and 1.87x speedup to tuned Adam on CIFAR10 and CIFAR100 respectively. In Figure 6 and Table 2, with the exception of PTB LSTM, momentum SGD also produces better training loss, as well as better validation perplexity in language modeling and better validation F1 in parsing. For the parsing task, we also compare with tuned Vanilla SGD and

Table 2: Speedup of YELLOWFIN and tuned mom. SGD over tuned Adam.

|  | Adam | mom.SGD | YF |
|---|---|---|---|
| CIFAR10 | 1x | 1.71x | 1.93x |
| CIFAR100 | 1x | 1.87x | 1.38x |
| PTB | 1x | 0.88x | 0.77x |
| TS | 1x | 2.49x | 3.28x |
| WSJ | 1x | 1.33x | 2.33x |

AdaGrad, which are used in the NLP community. Figure 6 (right) shows that *fixed momentum 0.9 can already speedup Vanilla SGD by* 2.73x, *achieving observably better validation F1*. We refer to Appendix J.2 for further discussion on the importance of momentum adaptivity in YELLOWFIN.

**YELLOWFIN can match hand-tuned momentum SGD and can outperform hand-tuned Adam** In our experiments, YELLOWFIN, without any hand-tuning, yields training loss matching hand-tuned momentum SGD for all the ResNet and LSTM models in Figure 6 and 9. When comparing to tuned Adam in Table 2, except being slightly slower on PTB LSTM, YELLOWFIN achieves 1.38x to 3.28x speedups in training losses on the other four models. *More importantly,* YELLOWFIN *consistently shows better validation metrics than tuned Adam in Figure 6*. It demonstrates that YELLOWFIN can match tuned momentum SGD and outperform tuned state-of-the-art adaptive optimizers. In Appendix J.4, we show YELLOWFIN further speeding up with finer-grain manual learning rate tuning.

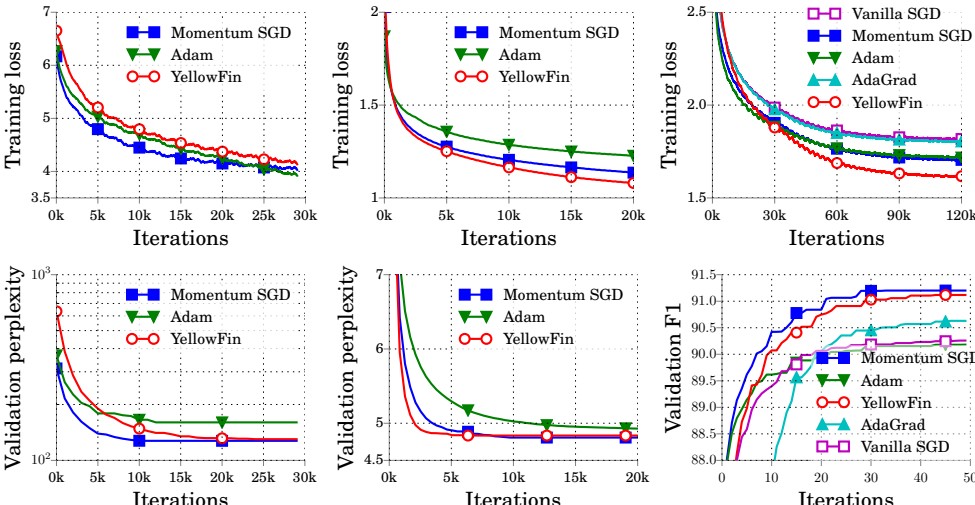

Figure 6: Training loss and test metrics on word-level language modeling with PTB (left), character-level language modeling with TS (middle) and constituency parsing on WSJ (right). Note the validation metrics are monotonic as we report the best values up to each specific number of iterations.

## 5.2 ASYNCHRONOUS EXPERIMENTS

In this section, we evaluate closed-loop YELLOWFIN with focus on the number of iterations to reach a certain solution. To that end, we run 16 asynchronous workers on a single machine and force them to update the model in a round-robin fashion, i.e. the gradient is delayed for 15 iterations. Figure 7 presents training losses on the CIFAR100 ResNet, using YELLOWFIN in Algorithm 1, closed-loop YELLOWFIN in Algorithm 5 and Adam with the learning rate achieving the best smoothed loss in Section 5.1. We can observe closed-loop YELLOWFIN achieves 20.1x speedup to YELLOWFIN, and consequently a 2.69x speedup to Adam. This demonstrates that (1) closed-loop YELLOWFIN accelerates by reducing algorithmic momentum to compensate for asynchrony and (2) can converge in less iterations than Adam in asynchronous-parallel training.

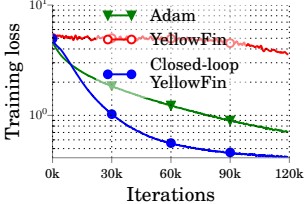

Figure 7: Asynchronous performance on CIFAR100 ResNet.

## 6 RELATED WORK

Many techniques have been proposed on tuning hyperparameters for optimizers. General hyper-parameter tuning approaches, such as random search (Bergstra and Bengio, 2012) and Bayesian approaches (Snoek et al., 2012; Hutter et al., 2011), directly applies to optimizer tuning. As another trend, adaptive methods, including AdaGrad (Duchi et al., 2011), RMSProp (Tieleman and Hinton, 2012) and Adam (Chilimbi et al., 2014), uses per-dimension learning rate. Schaul et al. (2013) use a noisy quadratic model similar to ours to extract learning rate tuning rule in Vanilla SGD. However they do not use momentum which is essential in training modern neural networks. Existing adaptive momentum approach either consider the deterministic setting (Graepel and Schraudolph, 2002; Rehman and Nawi, 2011; Hameed et al., 2016; Swanston et al., 1994; Ampazis and Perantonis, 2000; Qiu et al., 1992) or only analyze stochasticity with $O(1/t)$ learning rate (Leen and Orr, 1994). In the contrast, we aim at practical momentum adaptivity for stochastically training neural networks.

## 7 DISCUSSION

We presented YELLOWFIN, the first optimization method that automatically tunes momentum as well as the learning rate of momentum SGD. YELLOWFIN outperforms the state-of-the-art adaptive optimizers on a large class of models both in synchronous and asynchronous settings. It estimates statistics purely from the gradients of a running system, and then tunes the hyperparameters of momentum SGD based on noisy, local quadratic approximations. As future work, we believe that more accurate curvature estimation methods, like the *bbprop* method (Martens et al., 2012) can further improve YELLOWFIN. We also believe that our closed-loop momentum control mechanism in Section 4 could accelerate convergence for other adaptive methods in asynchronous-parallel settings.

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

## A    PROOF OF LEMMA 2

To prove Lemma 2, we first prove a more generalized version in Lemma 6. By restricting $f$ to be a one dimensional quadratics function, the generalized curvature $h_t$ itself is the only eigenvalue. We can prove Lemma 2 as a straight-forward corollary. Lemma 6 also implies, in the multiple dimensional correspondence of (4), the spectral radius $\rho(\boldsymbol{A}_t) = \sqrt{\mu}$ if the curvature on all eigenvector directions (eigenvalue) satisfies (5).

**Lemma 6.** *Let the gradients of a function $f$ be described by*

$$\nabla f(\boldsymbol{x}_t) = \boldsymbol{H}(\boldsymbol{x}_t)(\boldsymbol{x}_t - \boldsymbol{x}^*), \tag{15}$$

*with $\boldsymbol{H}(\boldsymbol{x}_t) \in \mathbb{R}^n \mapsto \mathbb{R}^{n \times n}$. Then the momentum update can be expressed as a linear operator:*

$$\begin{pmatrix} \boldsymbol{y}_{t+1} \\ \boldsymbol{y}_t \end{pmatrix} = \begin{pmatrix} \boldsymbol{I} - \alpha \boldsymbol{H}(\boldsymbol{x}_t) + \mu \boldsymbol{I} & -\mu \boldsymbol{I} \\ \boldsymbol{I} & \boldsymbol{0} \end{pmatrix} \begin{pmatrix} \boldsymbol{y}_t \\ \boldsymbol{y}_{t-1} \end{pmatrix} = \boldsymbol{A}_t \begin{pmatrix} \boldsymbol{y}_t \\ \boldsymbol{y}_{t-1} \end{pmatrix}, \tag{16}$$

*where $\boldsymbol{y}_t \triangleq \boldsymbol{x}_t - \boldsymbol{x}^*$. Now, assume that the following condition holds for all eigenvalues $\lambda(\boldsymbol{H}(\boldsymbol{x}_t))$ of $\boldsymbol{H}(\boldsymbol{x}_t)$:*

$$\frac{(1 - \sqrt{\mu})^2}{\alpha} \le \lambda(\boldsymbol{H}(\boldsymbol{x}_t)) \le \frac{(1 + \sqrt{\mu})^2}{\alpha}. \tag{17}$$

*then the spectral radius of $\boldsymbol{A}_t$ is controlled by momentum with $\rho(\boldsymbol{A}_t) = \sqrt{\mu}$.*

*Proof.* Let $\lambda_t$ be an eigenvalue of matrix $\boldsymbol{A}_t$, it gives $\det(\boldsymbol{A}_t - \lambda_t \boldsymbol{I}) = 0$. We define the blocks in $\boldsymbol{A}_t$ as $\boldsymbol{C} = \boldsymbol{I} - \alpha \boldsymbol{H}_t + \mu \boldsymbol{I} - \lambda_t \boldsymbol{I}$, $\boldsymbol{D} = -\mu \boldsymbol{I}$, $\boldsymbol{E} = \boldsymbol{I}$ and $\boldsymbol{F} = -\lambda_t \boldsymbol{I}$ which gives

$$\det(\boldsymbol{A}_t - \lambda_t \boldsymbol{I}) = \det \boldsymbol{F} \det\left(\boldsymbol{C} - \boldsymbol{D}\boldsymbol{F}^{-1}\boldsymbol{E}\right) = 0$$

assuming generally $\boldsymbol{F}$ is invertible. Note we use $\boldsymbol{H}_t \triangleq \boldsymbol{H}(\boldsymbol{x}_t)$ for simplicity in writing. The equation $\det\left(\boldsymbol{C} - \boldsymbol{D}\boldsymbol{F}^{-1}\boldsymbol{E}\right) = 0$ implies that

$$\det\left(\lambda_t^2 \boldsymbol{I} - \lambda_t \boldsymbol{M}_t + \mu \boldsymbol{I}\right) = 0 \tag{18}$$

with $\boldsymbol{M}_t = (\boldsymbol{I} - \alpha \boldsymbol{H}_t + \mu \boldsymbol{I})$. In other words, $\lambda_t$ satisfied that $\lambda_t^2 - \lambda_t \lambda(\boldsymbol{M}_t) + \mu = 0$ with $\lambda(\boldsymbol{M}_t)$ being one eigenvalue of $\boldsymbol{M}_t$. I.e.

$$\lambda_t = \frac{\lambda(\boldsymbol{M}_t) \pm \sqrt{\lambda(\boldsymbol{M}_t)^2 - 4\mu}}{2} \tag{19}$$

On the other hand, (17) guarantees that $(1 - \alpha\lambda(\boldsymbol{H}_t) + \mu)^2 \le 4\mu$. We know both $\boldsymbol{H}_t$ and $\boldsymbol{I} - \alpha \boldsymbol{H}_t + \mu \boldsymbol{I}$ are symmetric. Thus for all eigenvalues $\lambda(\boldsymbol{M}_t)$ of $\boldsymbol{M}_t$, we have $\lambda(\boldsymbol{M}_t)^2 = (1 - \alpha\lambda(\boldsymbol{H}_t) + \mu)^2 \le 4\mu$ which guarantees $|\lambda_t| = \sqrt{\mu}$ for all $\lambda_t$. As the spectral radius is equal to the magnitude of the largest eigenvalue of $\boldsymbol{A}_t$, we have the spectral radius of $\boldsymbol{A}_t$ being $\sqrt{\mu}$.

$\square$

## B    PROOF OF LEMMA 4

We first prove Lemma 7 and Lemma 8 as preparation for the proof of Lemma 4. After the proof for one dimensional case, we discuss the trivial generalization to multiple dimensional case.

**Lemma 7.** *Let the $h$ be the curvature of a one dimensional quadratic function $f$ and $\overline{x}_t = \mathbb{E}x_t$. We assume, without loss of generality, the optimum point of $f$ is $x^\star = 0$. Then we have the following recurrence*

$$\begin{pmatrix} \overline{x}_{t+1} \\ \overline{x}_t \end{pmatrix} = \begin{pmatrix} 1 - \alpha h + \mu & -\mu \\ 1 & 0 \end{pmatrix}^t \begin{pmatrix} x_1 \\ x_0 \end{pmatrix} \tag{20}$$

*Proof.* From the recurrence of momentum SGD, we have

$$\begin{aligned}
\mathbb{E}x_{t+1} &= \mathbb{E}[x_t - \alpha \nabla f_{S_t}(x_t) + \mu(x_t - x_{t-1})] \\
&= \mathbb{E}_{x_t}[x_t - \alpha \mathbb{E}_{S_t} \nabla f_{S_t}(x_t) + \mu(x_t - x_{t-1})] \\
&= \mathbb{E}_{x_t}[x_t - \alpha h x_t + \mu(x_t - x_{t-1})] \\
&= (1 - \alpha h + \mu)\overline{x}_t - \mu\overline{x}_{t-1}
\end{aligned}$$

By putting the equation in to matrix form, (20) is a straight-forward result from unrolling the recurrence for $t$ times. Note as we set $x_1 = x_0$ with no uncertainty in momentum SGD, we have $[\overline{x}_0, \overline{x}_1] = [x_0, x_1]$. $\qquad\square$

**Lemma 8.** *Let $U_t = \mathbb{E}(x_t - \overline{x}_t)^2$ and $V_t = \mathbb{E}(x_t - \overline{x}_t)(x_{t-1} - \overline{x}_{t-1})$ with $\overline{x}_t$ being the expectation of $x_t$. For quadratic function $f(x)$ with curvature $h \in \mathbb{R}$, We have the following recurrence*

$$\begin{pmatrix} U_{t+1} \\ U_t \\ V_{t+1} \end{pmatrix} = (\boldsymbol{I} - \boldsymbol{B}^\top)(\boldsymbol{I} - \boldsymbol{B})^{-1} \begin{pmatrix} \alpha^2 C \\ 0 \\ 0 \end{pmatrix} \tag{21}$$

*where*

$$\boldsymbol{B} = \begin{pmatrix} (1 - \alpha h + \mu)^2 & \mu^2 & -2\mu(1 - \alpha h + \mu) \\ 1 & 0 & 0 \\ 1 - \alpha h + \mu & 0 & -\mu \end{pmatrix} \tag{22}$$

*and $C = \mathbb{E}(\nabla f_{S_t}(x_t) - \nabla f(x_t))^2$ is the variance of gradient on minibatch $S_t$.*

*Proof.* We prove by first deriving the recurrence for $U_t$ and $V_t$ respectively and combining them in to a matrix form. For $U_t$, we have

$$\begin{aligned}
U_{t+1} &= \mathbb{E}(x_{t+1} - \overline{x}_{t+1})^2 \\
&= \mathbb{E}(x_t - \alpha \nabla f_{S_t}(x_t) + \mu(x_t - x_{t-1}) - (1 - \alpha h + \mu)\overline{x}_t + \mu\overline{x}_{t-1})^2 \\
&= \mathbb{E}(x_t - \alpha \nabla f(x_t) + \mu(x_t - x_{t-1}) - (1 - \alpha h + \mu)\overline{x}_t + \mu\overline{x}_{t-1} + \alpha(\nabla f(x_t) - \nabla f_{S_t}(x_t)))^2 \\
&= \mathbb{E}((1 - \alpha h + \mu)(x_t - \overline{x}_t) - \mu(x_{t-1} - \overline{x}_{t-1}))^2 + \alpha^2 \mathbb{E}(\nabla f(x_t) - \nabla f_{S_t}(x_t))^2 \\
&= (1 - \alpha h + \mu)^2 \mathbb{E}(x_t - \overline{x}_t)^2 - 2\mu(1 - \alpha h + \mu)\mathbb{E}(x_t - \overline{x}_t)(x_{t-1} - \overline{x}_{t-1}) \\
&\quad + \mu^2 \mathbb{E}(x_{t-1} - \overline{x}_{t-1})^2 + \alpha^2 C
\end{aligned} \tag{23}$$

where the cross terms cancels due to the fact $\mathbb{E}_{S_t}[\nabla f(x_t) - \nabla f_{S_t}(x_t)] = 0$ in the third equality.

For $V_t$, we can similarly derive

$$\begin{aligned}
V_t &= \mathbb{E}(x_t - \overline{x}_t)(x_{t-1} - \overline{x}_{t-1}) \\
&= \mathbb{E}((1 - \alpha h + \mu)(x_{t-1} - \overline{x}_{t-1}) - \mu(x_{t-2} - \overline{x}_{t-2}) + \alpha(\nabla f(x_t) - \nabla f_{S_t}(x_t)))(x_{t-1} - \overline{x}_{t-1}) \\
&= (1 - \alpha h + \mu)\mathbb{E}(x_{t-1} - \overline{x}_{t-1})^2 - \mu\mathbb{E}(x_{t-1} - \overline{x}_{t-1})(x_{t-2} - \overline{x}_{t-2})
\end{aligned} \tag{24}$$

Again, the term involving $\nabla f(x_t) - \nabla f_{S_t}(x_t)$ cancels in the third equality as a results of $\mathbb{E}_{S_t}[\nabla f(x_t) - \nabla f_{S_t}(x_t)] = 0$. (23) and (24) can be jointly expressed in the following matrix form

$$\begin{pmatrix} U_{t+1} \\ U_t \\ V_{t+1} \end{pmatrix} = \boldsymbol{B} \begin{pmatrix} U_t \\ U_{t-1} \\ V_t \end{pmatrix} + \begin{pmatrix} \alpha^2 C \\ 0 \\ 0 \end{pmatrix} = \sum_{i=0}^{t-1} \boldsymbol{B}^i \begin{pmatrix} \alpha^2 C \\ 0 \\ 0 \end{pmatrix} + \boldsymbol{B}^t \begin{pmatrix} U_1 \\ U_0 \\ V_1 \end{pmatrix} = (\boldsymbol{I} - \boldsymbol{B}^t)(\boldsymbol{I} - \boldsymbol{B})^{-1} \begin{pmatrix} \alpha^2 C \\ 0 \\ 0 \end{pmatrix}. \tag{25}$$

Note the second term in the second equality is zero because $x_0$ and $x_1$ are deterministic. Thus $U_1 = U_0 = V_1 = 0$. $\qquad\square$

According to Lemma 7 and 8, we have $\mathbb{E}(\overline{x}_t - x^*)^2 = (\boldsymbol{e}_1^\top \boldsymbol{A}^t [x_1, x_0]^\top)^2$ and $\mathbb{E}(x_t - \overline{x}_t)^2 = \alpha^2 C \boldsymbol{e}_1^\top (\boldsymbol{I} - \boldsymbol{B}^t)(\boldsymbol{I} - \boldsymbol{B})^{-1} \boldsymbol{e}_1$ where $\boldsymbol{e}_1 \in \mathbb{R}^n$ has all zero entries but the first dimension. Combining these two terms, we prove Lemma 4. Though the proof here is for one dimensional quadratics, it trivially generalizes to multiple dimensional quadratics. Specifically, we can decompose the quadratics along the eigenvector directions, and then apply Lemma 4 to each eigenvector direction using the corresponding curvature $h$ (eigenvalue). By summing quantities in (10) for all eigenvector directions, we can achieve the multiple dimensional correspondence of (10).

## C   PROOF OF LEMMA 5

Again we first present a proof of a multiple dimensional generalized version of Lemma 5. The proof of Lemma 5 is a one dimensional special case of Lemma 9. Lemma 9 also implies that for multiple dimension quadratics, the corresponding spectral radius $\rho(\boldsymbol{B}) = \mu$ if $\frac{(1-\sqrt{\mu})^2}{\alpha} \leq h \leq \frac{(1+\sqrt{\mu})^2}{\alpha}$ on all the eigenvector directions with $h$ being the eigenvalue (curvature).

**Lemma 9.** *Let $\boldsymbol{H} \in \mathbb{R}^{n \times n}$ be a symmetric matrix and $\rho(\boldsymbol{B})$ be the spectral radius of matrix*

$$\boldsymbol{B} = \begin{pmatrix} (\boldsymbol{I} - \alpha\boldsymbol{H} + \mu\boldsymbol{I})^\top(\boldsymbol{I} - \alpha\boldsymbol{H} + \mu\boldsymbol{I}) & \mu^2\boldsymbol{I} & -2\mu(\boldsymbol{I} - \alpha\boldsymbol{H} + \mu\boldsymbol{I}) \\ \boldsymbol{I} & \boldsymbol{0} & \boldsymbol{0} \\ \boldsymbol{I} - \alpha\boldsymbol{H} + \mu\boldsymbol{I} & \boldsymbol{0} & -\mu\boldsymbol{I} \end{pmatrix} \tag{26}$$

*We have $\rho(\boldsymbol{B}) = \mu$ if all eigenvalues $\lambda(\boldsymbol{H})$ of $\boldsymbol{H}$ satisfies*

$$\frac{(1 - \sqrt{\mu})^2}{\alpha} \leq \lambda(\boldsymbol{H}) \leq \frac{(1 + \sqrt{\mu})^2}{\alpha}. \tag{27}$$

*Proof.* Let $\lambda$ be an eigenvalue of matrix $\boldsymbol{B}$, it gives $\det(\boldsymbol{B} - \lambda\boldsymbol{I}) = 0$ which can be alternatively expressed as

$$\det(\boldsymbol{B} - \lambda\boldsymbol{I}) = \det\boldsymbol{F}\det(\boldsymbol{C} - \boldsymbol{D}\boldsymbol{F}^{-1}\boldsymbol{E}) = 0 \tag{28}$$

assuming $\boldsymbol{F}$ is invertible, i.e. $\lambda + \mu \neq 0$, where the blocks in $\boldsymbol{B}$

$$\boldsymbol{C} = \begin{pmatrix} \boldsymbol{M}^\top\boldsymbol{M} - \lambda\boldsymbol{I} & \mu^2\boldsymbol{I} \\ \boldsymbol{I} & -\lambda\boldsymbol{I} \end{pmatrix}, \boldsymbol{D} = \begin{pmatrix} -2\mu\boldsymbol{M} \\ \boldsymbol{0} \end{pmatrix}, \boldsymbol{E} = \begin{pmatrix} \boldsymbol{M} \\ \boldsymbol{0} \end{pmatrix}^\top, \boldsymbol{F} = -\mu\boldsymbol{I} - \lambda\boldsymbol{I}$$

with $\boldsymbol{M} = \boldsymbol{I} - \alpha\boldsymbol{H} + \mu\boldsymbol{I}$. (28) can be transformed using straight-forward algebra as

$$\det\begin{pmatrix} (\lambda - \mu)\boldsymbol{M}^\top\boldsymbol{M} - (\lambda + \mu)\lambda\boldsymbol{I} & (\lambda + \mu)\mu^2\boldsymbol{I} \\ (\lambda + \mu)\boldsymbol{I} & -(\lambda + \mu)\lambda\boldsymbol{I} \end{pmatrix} = 0 \tag{29}$$

Using similar simplification technique as in (28), we can further simplify into

$$(\lambda - \mu)\det\left((\lambda + \mu)^2\boldsymbol{I} - \lambda\boldsymbol{M}^\top\boldsymbol{M}\right) = 0 \tag{30}$$

if $\lambda \neq \mu$, as $(\lambda + \mu)^2\boldsymbol{I} - \lambda\boldsymbol{M}^\top\boldsymbol{M}$ is diagonalizable, we have $(\lambda + \mu)^2 - \lambda\lambda(\boldsymbol{M})^2 = 0$ with $\lambda(\boldsymbol{M})$ being an eigenvalue of symmetric $\boldsymbol{M}$. The analytic solution to the equation can be explicitly expressed as

$$\lambda = \frac{\lambda(\boldsymbol{M})^2 - 2\mu \pm \sqrt{(\lambda(\boldsymbol{M})^2 - 2\mu)^2 - 4\mu^2}}{2}. \tag{31}$$

When the condition in (27) holds, we have $\lambda(\boldsymbol{M})^2 = (1 - \alpha\lambda(\boldsymbol{H}) + \mu)^2 \leq 4\mu$. One can verify that

$$\begin{aligned} (\lambda(\boldsymbol{M})^2 - 2\mu)^2 - 4\mu^2 &= (\lambda(\boldsymbol{M})^2 - 4\mu)\lambda(\boldsymbol{M})^2 \\ &= \left((1 - \alpha\rho(\boldsymbol{H}) + \mu)^2 - 4\mu\right)\lambda(\boldsymbol{M})^2 \\ &\leq 0 \end{aligned} \tag{32}$$

Thus the roots in (31) are conjugate with $|\lambda| = \mu$. In conclusion, the condition in (27) can guarantee all the eigenvalues of $\boldsymbol{B}$ has magnitude $\mu$. Thus the spectral radius of $\boldsymbol{B}$ is controlled by $\mu$.   $\square$

## D   ANALYTICAL SOLUTION TO (14)

The problem in (14) does not need iterative solver but has an analytical solution. Substituting only the second constraint, the objective becomes $p(x) = x^2D^2 + (1 - x)^4/h_{\min}^2C$ with $x = \sqrt{\mu} \in [0, 1)$. By setting the gradient of $p(x)$ to 0, we can get a cubic equation whose root $x = \sqrt{\mu_p}$ can be computed in closed form using Vieta's substitution. As $p(x)$ is uni-modal in $[0, 1)$, the optimizer for (14) is exactly the maximum of $\mu_p$ and $(\sqrt{h_{\max}/h_{\min}} - 1)^2/(\sqrt{h_{\max}/h_{\min}} + 1)^2$, the right hand-side of the first constraint in (14).

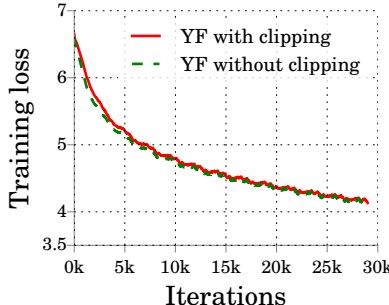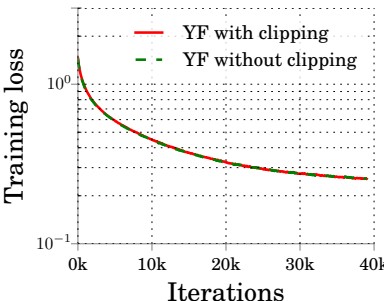

Figure 8: Training losses on PTB LSTM (left) and CIFAR10 ResNet (right) for YellowFin with and without adaptive clipping.

## E  PRACTICAL IMPLEMENTATION

In Section 3.2, we discuss estimators for learning rate and momentum tuning in YELLOWFIN. In our experiment practice, we have identified a few practical implementation details which are important for improving estimators. Zero-debias is proposed by Kingma and Ba (2014), which accelerates the process where exponential average adapts to the level of original quantity in the beginning. We applied zero-debias to all the exponential average quantities involved in our estimators. In some LSTM models, we observe that our estimated curvature may decrease quickly along the optimization process. In order to better estimate extremal curvature $h_{\max}$ and $h_{\min}$ with fast decreasing trend, we apply zero-debias exponential average on the logarithmic of $h_{\max,t}$ and $h_{\min,t}$, instead of directly on $h_{\max,t}$ and $h_{\min,t}$. Except from the above two techniques, we also implemented the slow start heuristic proposed by (Schaul et al., 2013). More specifically, we use $\alpha = \min\{\alpha_t, t \cdot \alpha_t/(10 \cdot w)\}$ as our learning rate with $w$ as the size of our sliding window in $h_{\max}$ and $h_{\min}$ estimation. It discount the learning rate in the first $10 \cdot w$ steps and helps to keep the learning rate small in the beginning when the exponential averaged quantities are not accurate enough.

## F  ADAPTIVE GRADIENT CLIPPING IN YELLOWFIN

Gradient clipping has been established in literature as a standard—almost necessary—tool for training such objectives (Pascanu et al., 2013; Goodfellow et al., 2016; Gehring et al., 2017). However, the classic tradeoff between adaptivity and stability applies: setting a clipping threshold that is too low can hurt performance; setting it to be high, can compromise stability. YELLOWFIN, keeps running estimates of extremal gradient magnitude squares, $h_{max}$ and $h_{min}$ in order to estimate a generalized condition number. We posit that $\sqrt{h_{max}}$ is an ideal gradient norm threshold for adaptive clipping. In order to ensure robustness to extreme gradient spikes, like the ones in Figure 4, we also limit the growth rate of the envelope $h_{max}$ in Algorithm 2 as follows:

$$h_{max} \leftarrow \beta \cdot h_{max} + (1 - \beta) \cdot \min\{h_{max,t}, 100 \cdot h_{max}\} \tag{33}$$

Our heuristics follows along the lines of classic recipes like Pascanu et al. (2013). However, instead of using the average gradient norm to clip, it uses a running estimate of the maximum norm $h_{\max}$.

In Section 3.3, we saw that adaptive clipping stabilizes the training on objectives that exhibit exploding gradients. In Figure 8, we demonstrate that the adaptive clipping does not hurt performance on models that do not exhibit instabilities without clipping. Specifically, for both PTB LSTM and CIFAR10 ResNet, the difference between YELLOWFIN with and without adaptive clipping diminishes quickly.

## G  CLOSED-LOOP YELLOWFIN FOR ASYNCHRONOUS TRAINING

In Section 4, we briefly discuss the closed-loop momentum control mechanism in closed-loop YELLOWFIN. In this section, after presenting more preliminaries on asynchrony, we show with

details on the mechanism: it measures the dynamics on a running system and controls momentum with a negative feedback loop.

**Preliminaries** Asynchrony is a popular parallelization technique (Niu et al., 2011) that avoids synchronization barriers. When training on $M$ asynchronous workers, staleness (the number of model updates between a worker's read and write operations) is on average $\tau = M - 1$, i.e., the gradient in the SGD update is delayed by $\tau$ iterations as $\nabla f_{S_{t-\tau}}(x_{t-\tau})$. Asynchrony yields faster steps, but can increase the number of iterations to achieve the same solution, a tradeoff between hardware and statistical efficiency (Zhang and Ré, 2014). Mitliagkas et al. (2016) interpret asynchrony as added momentum dynamics. Experiments in Hadjis et al. (2016) support this finding, and demonstrate that reducing algorithmic momentum can compensate for asynchrony-induced momentum and significantly reduce the number of iterations for convergence. Motivated by that result, we use the model in (34), where the total momentum, $\mu_T$, includes both asynchrony-induced and algorithmic momentum, $\mu$, in (1).

$$\mathbb{E}[x_{t+1} - x_t] = \mu_T \mathbb{E}[x_t - x_{t-1}] - \alpha \mathbb{E}\nabla f(x_t) \tag{34}$$

We will use this expression to design an estimator for the value of total momentum, $\hat{\mu}_T$. This estimator is a basic building block of closed-loop YELLOWFIN, that *removes the need to manually compensate for the effects of asynchrony*.

**Measuring the momentum dynamics** Closed-loop YELLOWFIN estimates total momentum $\mu_T$ on a running system and uses a negative feedback loop to adjust algorithmic momentum accordingly. Equation (14) gives an estimate of $\hat{\mu}_T$ on a system with staleness $\tau$, based on (14).

$$\hat{\mu}_T = \mathsf{median}\left( \frac{x_{t-\tau} - x_{t-\tau-1} + \alpha \nabla_{S_{t-\tau-1}} f(x_{t-\tau-1})}{x_{t-\tau-1} - x_{t-\tau-2}} \right) \tag{35}$$

We use $\tau$-stale model values to match the staleness of the gradient, and perform all operations in an elementwise fashion. This way we get a total momentum measurement from each variable; the median combines them into a more robust estimate.

**Closing the asynchrony loop** Given a reliable measurement of $\mu_T$, we can use it to adjust the value of algorithmic momentum so that the total momentum matches the *target momentum* as decided by YELLOWFIN in Algorithm 1. Closed-loop YELLOWFIN in Algorithm 5 uses a simple negative feedback loop to achieve the adjustment.

---

**Algorithm 5** Closed-loop YELLOWFIN

1: Input: $\mu \leftarrow 0$, $\alpha \leftarrow 0.0001$, $\gamma \leftarrow 0.01$, $\tau$ (staleness)
2: **for** $t \leftarrow 1$ to $T$ **do**
3:      $x_t \leftarrow x_{t-1} + \mu(x_{t-1} - x_{t-2}) - \alpha \nabla_{S_t} f(x_{t-\tau-1})$
4:      $\mu^*, \alpha \leftarrow$ YELLOWFIN$(\nabla_{S_t} f(x_{t-\tau-1}), \beta)$
5:      $\hat{\mu}_T \leftarrow \mathsf{median}\left( \frac{x_{t-\tau} - x_{t-\tau-1} + \alpha \nabla_{S_{t-\tau-1}} f(x_{t-\tau-1})}{x_{t-\tau-1} - x_{t-\tau-2}} \right)$      ▷ Measuring total momentum
6:      $\mu \leftarrow \mu + \gamma \cdot (\mu^* - \hat{\mu}_T)$      ▷ Closing the loop
7: **end for**

---

## H   Model specification

The model specification is shown in Table 3 for all the experiments in Section 5. CIRAR10 ResNet uses the regular ResNet units while CIFAR100 ResNet uses the bottleneck units. Only the convolutional layers are shown with filter size, filter number as well as the repeating count of the units. The layer counting for ResNets also includes batch normalization and Relu layers. The LSTM models are also diversified for different tasks with different vocabulary sizes, word embedding dimensions and number of layers.

## I   Specification for synchronous experiments

In Section 5.1, we demonstrate the synchronous experiments with extensive discussions. For the reproducibility, we provide here the specification of learning rate grids. The number of iterations as

Table 3: Specification of ResNet and LSTM model architectures.

| network | # layers | Conv 0 | Unit 1s | Unit 2s | Unit 3s |
|---|---|---|---|---|---|
| CIFAR10 ResNet | 110 | $\begin{bmatrix} 3 \times 3, & 4 \end{bmatrix}$ | $\begin{bmatrix} 3 \times 3, & 4 \\ 3 \times 3, & 4 \end{bmatrix} \times 6$ | $\begin{bmatrix} 3 \times 3, & 8 \\ 3 \times 3, & 8 \end{bmatrix} \times 6$ | $\begin{bmatrix} 3 \times 3, & 16 \\ 3 \times 3, & 16 \end{bmatrix} \times 6$ |
| CIFAR100 ResNet | 164 | $\begin{bmatrix} 3 \times 3, & 4 \end{bmatrix}$ | $\begin{bmatrix} 1 \times 1, & 16 \\ 3 \times 3, & 16 \\ 1 \times 1, & 64 \end{bmatrix} \times 6$ | $\begin{bmatrix} 1 \times 1, & 32 \\ 3 \times 3, & 32 \\ 1 \times 1, & 128 \end{bmatrix} \times 6$ | $\begin{bmatrix} 1 \times 1, & 64 \\ 3 \times 3, & 64 \\ 1 \times 1, & 256 \end{bmatrix} \times 6$ |
| network | # layers | Word Embed. | Layer 1 | Layer 2 | Layer 3 |
| TS LSTM | 2 | [65 vocab, 128 dim] | 128 hidden units | 128 hidden units | – |
| PTB LSTM | 2 | [10000 vocab, 200 dim] | 200 hidden units | 200 hidden units | – |
| WSJ LSTM | 3 | [6922 vocab, 500 dim] | 500 hidden units | 500 hidden units | 500 hidden units |

well as epochs, i.e. the number of passes over the full training sets, are also listed for completeness. For YELLOWFIN in all the experiments in Section 5, we uniformly use sliding window size 20 for extremal curvature estimation and $\beta = 0.999$ for smoothing. For momentum SGD and Adam, we use the following configurations.

- CIFAR10 ResNet
  - 40k iterations ($\sim$114 epochs)
  - Momentum SGD learning rates $\{0.001, 0.01(\text{best}), 0.1, 1.0\}$, momentum 0.9
  - Adam learning rates $\{0.0001, 0.001(\text{best}), 0.01, 0.1\}$
- CIFAR100 ResNet
  - 120k iterations ($\sim$341 epochs)
  - Momentum SGD learning rates $\{0.001, 0.01(\text{best}), 0.1, 1.0\}$, momentum 0.9
  - Adam learning rates $\{0.00001, 0.0001(\text{best}), 0.001, 0.01\}$
- PTB LSTM
  - 30k iterations ($\sim$13 epochs)
  - Momentum SGD learning rates $\{0.01, 0.1, 1.0(\text{best}), 10.0\}$, momentum 0.9
  - Adam learning rates $\{0.0001, 0.001(\text{best}), 0.01, 0.1\}$
- TS LSTM
  - $\sim$21k iterations (50 epochs)
  - Momentum SGD learning rates $\{0.05, 0.1, 0.5, 1.0(\text{best}), 5.0\}$, momentum 0.9
  - Adam learning rates $\{0.0005, 0.001, 0.005(\text{best}), 0.01, 0.05\}$
  - Decrease learning rate by factor 0.97 every epoch for all optimizers, following the design by Karpathy et al. (2015).
- WSJ LSTM
  - $\sim$120k iterations (50 epochs)
  - Momentum SGD learning rates $\{0.05, 0.1, 0.5(\text{best}), 1.0, 5.0\}$, momentum 0.9
  - Adam learning rates $\{0.0001, 0.0005, 0.001(\text{best}), 0.005, 0.01\}$
  - Vanilla SGD learning rates $\{0.05, 0.1, 0.5, 1.0(\text{best}), 5.0\}$
  - Adagrad learning rates $\{0.05, 0.1, 0.5(\text{best}), 1.0, 5.0\}$
  - Decrease learning rate by factor 0.9 every epochs after 14 epochs for all optimizers, following the design by Choe and Charniak.

## J ADDITIONAL EXPERIMENT RESULTS

### J.1 TRAINING LOSSES ON CIFAR10 AND CIFAR100 RESNET

In Figure 9, we demonstrate the training loss on CIFAR10 ResNet and CIFAR100 ResNet. Specifically, YELLOWFIN can match the performance of hand-tuned momentum SGD, and achieves 1.93x and 1.38x speedup comparing to hand-tuned Adam respectively on CIFAR10 and CIFAR100 ResNet.

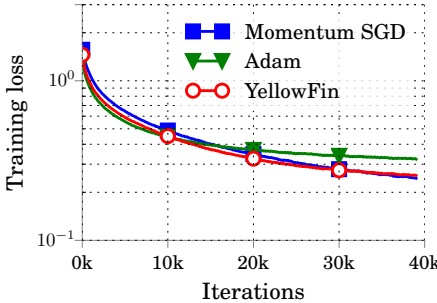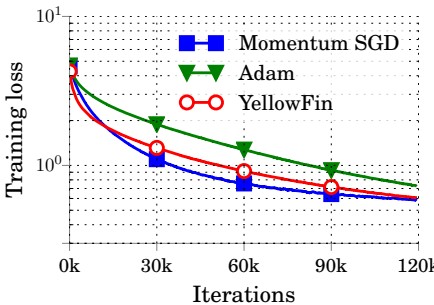

Figure 9: Training loss for ResNet on 100-layer CIFAR10 ResNet (left) and 164-layer CIFAR100 bottleneck ResNet.

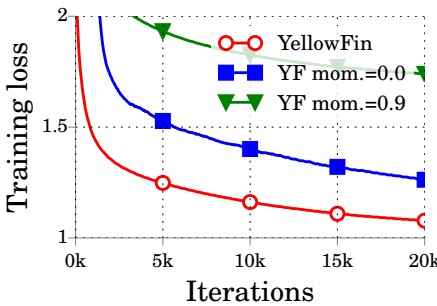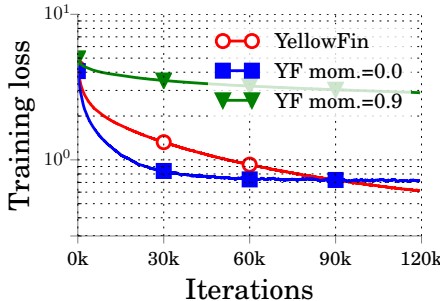

Figure 10: Training loss comparison between YELLOWFIN with adaptive momentum and YEL-LOWFIN with fixed momentum value.

## J.2 IMPORTANCE OF MOMENTUM ADAPTIVITY

To further emphasize the importance of momentum adaptivity in YELLOWFIN, we run YF on CIFAR100 ResNet and TS LSTM. In the experiments, YELLOWFIN tunes the learning rate. Instead of also using the momentum tuned by YF, we continuously feed prescribed momentum value 0.0 and 0.9 to the underlying momentum SGD optimizer which YF is tuning. In Figure 10, when comparing to YELLOWFIN with prescribed momentum 0.0 or 0.9, YELLOWFIN with adaptively tuned momentum achieves observably faster convergence on both TS LSTM and CIFAR100 ResNet. It empirically demonstrates the essential role of momentum adaptivity in YELLOWFIN.

## J.3 TUNING MOMENTUM CAN IMPROVE ADAM IN ASYNCHRONOUS-PARALLEL SETTING

We conduct experiments on PTB LSTM with 16 asynchronous workers using Adam using the same protocol as in Section 5.2. Fixing the learning rate to the value achieving the lowest smoothed loss in Section 5.1, we sweep the smoothing parameter $\beta_1$ (Kingma and Ba, 2014) of the first order moment estimate in grid $\{-0.2, 0.0, 0.3, 0.5, 0.7, 0.9\}$. $\beta_1$ serves the same role as momentum in SGD and we call it the momentum in Adam. Figure 11 shows tuning momentum for Adam under asynchrony gives measurably better training loss. This result emphasizes the importance of momentum tuning in asynchronous settings and suggests that state-of-the-art adaptive methods can perform sub-optimally when using prescribed momentum.

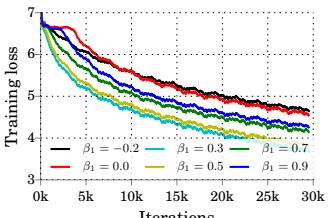

Figure 11: Hand-tuning Adam's momentum under asynchrony.

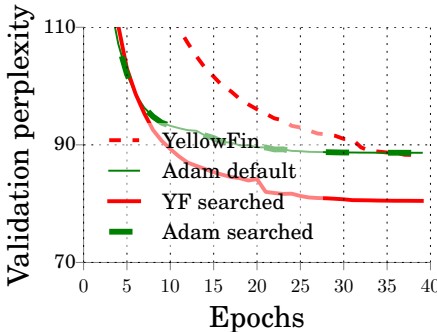 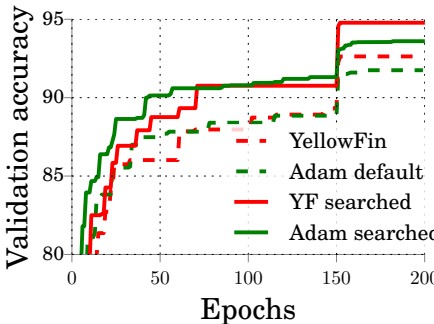

Figure 12: Validation perplexity on Tied LSTM and validation accuracy on ResNext. Learning rate fine-tuning using grid-searched factor can further improve the performance of YELLOWFIN in Algorithm 1. YELLOWFIN with learning factor search can outperform hand-tuned Adam on validation metrics on both models.

### J.4 ACCELERATING YELLOWFIN WITH FINER GRAIN LEARNING RATE TUNING

As an adaptive tuner, YELLOWFIN does not involve manual tuning. It can present faster development iterations on model architectures than grid search on optimizer hyperparameters. In deep learning practice for computer vision and natural language processing, after fixing the model architecture, extensive optimizer tuning (e.g. grid search or random search) can further improve the performance of a model. A natural question to ask is can we also slightly tune YELLOWFIN to accelerate convergence and improve the model performance. Specifically, we can manually multiply a positive number, the learning rate factor, to the auto-tuned learning rate in YELLOWFIN to further accelerate.

In this section, we empirically demonstrate the effectiveness of learning rate factor on a 29-layer ResNext (2x64d) (Xie et al., 2016) on CIFAR10 and a Tied LSTM model (Press and Wolf, 2016) with 650 dimensions for word embedding and two hidden units layers on the PTB dataset. When running YELLOWFIN, we search for the optimal learning rate factor in grid $\{\frac{1}{3}, 0.5, 1, 2(\text{best for ResNext}), 3(\text{best for Tied LSTM}), 10\}$. Similarly, we search the same learning rate factor grid for Adam, multiplying the factor to its default learning rate $0.001$. To further strength the performance of Adam as a baseline, we also run it on conventional logarithmic learning rate grid $\{5e^{-5}, 1e^{-4}, 5e^{-4}, 1e^{-3}, 5e^{-3}\}$ for ResNext and $\{1e^{-4}, 5e^{-4}, 1e^{-3}, 5e^{-3}, 1e^{-2}\}$ for Tied LSTM. We report the best metric from searching the union of learning rate factor grid and logarithmic learning rate grid as searched Adam results. Empirically, learning factor $\frac{1}{3}$ and $1.0$ works best for Adam respectively on ResNext and Tied LSTM.

As shown in Figure 12, with the searched best learning rate factor, YELLOWFIN can improve validation perplexity on Tied LSTM from $88.7$ to $80.5$, an improvement of more than $9\%$. Similarly, the searched learning rate factor can improve test accuracy from $92.63$ to $94.75$ on ResNext. More importantly, we can observe, with learning rate factor search on the two models, YELLOWFIN can achieve better validation metric than the searched Adam results. It demonstrates that finer-grain learning rate tuning, i.e. the learning rate factor search, can be effectively applied on YELLOWFIN to improve the performance of deep learning models.

