# OpenReview forum: "YellowFin and the Art of Momentum Tuning"
_ICLR.cc/2018/Conference — Reject_

### Official Review · AnonReviewer1 · 2017-11-27
**YellowFin and the Art of Momentum Tuning**

**Rating:** 4
**Confidence:** 3

**Review:**

This paper proposes a method to automatically tuning the momentum parameter in momentum SGD methods, which achieves better results and fast convergence speed than state-of-the-art Adam algorithm.

Although the results are promising, I found the presentation of this paper almost inaccessible to me.

First, though a minor point, but where does the name *YellowFin* come from?

For the presentation, the motivation in introduction is fine, but the following section about momentum operator is hard to follow. There are a lot of undefined notation. For example, what does the *convergence rate* mean (what is the measurement for convergence)? And is the *optimal accelerated rate* the same as *convergence rate* mentioned above? Also, what do you mean by *all directions* in the sentence below eq.2?

Then the paper talks about robustness properties of the momentum operator. But: first, I am not sure why the derivative of f(x) is defined as in eq.3, how is that related to the original definition of derivative?

In the following paragraph, what is *contraction*? Does it have anything to do with the paper as I didn't see it in the remaining text?

Lemma 2 seems to use the spectral radius of the momentum operator as the *robustness*. But how can it describe the robustness? More details are needed to understand this.

What it comes to Section 3, it seems to me that the authors try to use a local quadratic approximation for the original function f(x), and use the results in last section to find the optimal momentum parameter. I got confused in this section because eq.9 defines f(x) as a quadratic function. Is this f(x) the original function (non quadratic) or just the local quadratic approximation? If it is the local quadratic approximation, how is it correlated to the original function? It seems to me that the authors try to say if h and C are calculated from the original function, then this f(x) is a local quadratic approximation? If what I think is correct, I think it would be important to show this.

Also, the objective function in SingleStep algorithm seems to come from eq.13, but I failed to get the exact reasoning.

Overall, I think this is an interesting paper, but the presentation is too fuzzy to get it evaluated.

---

> ### Public Comment · (anonymous) · 2017-12-20
> **Part II: On the noisy quadratic model and the objective function in SingleStep algorithm**
>
> Q: What it comes to Section 3, it seems to me that the authors try to use a local quadratic approximation for the original function f(x), and use the results in last section to find the optimal momentum parameter. I got confused in this section because eq.9 defines f(x) as a quadratic function. Is this f(x) the original function (non quadratic) or just the local quadratic approximation? If it is the local quadratic approximation, how is it correlated to the original function? It seems to me that the authors try to say if h and C are calculated from the original function, then this f(x) is a local quadratic approximation? If what I think is correct, I think it would be important to show this.
>
> f(x) is the quadratic approximation of the original function. As AnonReviewer1 pointed out, we measure h and C from the original function. The h and C measurements are used to construct the local quadratic approximation and fed into the tuning rule. We would rephrase the statement on f(x) to connect it to the original function.
>
>
> Q: The objective function in SingleStep algorithm seems to come from eq.13, but I failed to get the exact reasoning.
>
> The SingleStep objective is a generalization of Equ (13) from 1D quadratics to multidimensional local quadratic approximations. Specifically, the SingleStep objective is the expected squared distance to the optimum of multiple dimensional local quadratic approximation after a single iterative step. For a multidimensional quadratic aligned with the axes, as we use *a single global learning rate and a single global momentum for the whole model*, the objective can be decomposed into sum of expected squared distance along different axes (i.e. on 1d quadratic), which is the left-hand side of Equ (13) (with t = 1 in Equ (13)). Note if the quadratic function is not axes-aligned, we can still decompose along the eigendirections of the Hessian instead of the axes.

---

> ### Public Comment · (anonymous) · 2017-12-20
> **Part I: On convergence rate, contraction and robustness of momentum operator**
>
> We appreciate AnonReviewer1's helpful comments about improving clarity. *We will upload a new manuscripts, with the suggestions and comments on clarification incorporated, in the next couple of days.* We answer the questions below inline. If there are any further ones, we would be happy to discuss and use them to keep improving the manuscript's clarity.
>
> Q: Where does the name *YellowFin* come from?
>
> We wanted a mnemonic for our ‘tuner’. Yellowfin happens to be one of the fastest species of tuna.
>
>
> Q: What does the *convergence rate* mean (what is the measurement for convergence)? And is the *optimal accelerated rate* the same as *convergence rate* mentioned above? Also, what do you mean by *all directions* in the sentence below eq.2?
>
>
> The convergence rate is with respect to the distance to the optimum. Specifically, the rate is \beta if and only if \|x_t - x* \| <= \beta^t \| x_0 - x*  \|, where x* is the optimum and x_t is the point after t steps in an iterative optimization process. The *optimal accelerated rate* is also with respect to distance towards the optimum, i.e. the optimal \beta in the above definition of convergence rate.
>
> Our statement on “all directions” is in the context of multidimensional quadratics with different curvature on different axes (i.e. \kappa > 1). Specifically, with large enough momentum \mu and proper learning rate, momentum gradient descent has the same convergence rate \sqrt{\mu} along all the axes, i.e. | x_{i, t} - x* | <= \sqrt{\mu}^t | x_{i, 0} - x*| where x_{i, t} is the coordinate on axis i after t steps. This holds even if the eigendirection of the quadratics are not axes-aligned. We will rephrase in the updated version to clarify.
>
>
> Q: Then the paper talks about robustness properties of the momentum operator. But: first, I am not sure why the derivative of f(x) is defined as in eq.3, how is that related to the original definition of derivative?
>
> Definition 1 is not to re-define derivative. Instead, we define generalized curvature h  for 1d functions and rewrite the derivative in terms of h. We will rephrase to clarify in the manuscript.
>
>
> Q: In the following paragraph, what is *contraction*? Does it have anything to do with the paper as I didn't see it in the remaining text?
>
> The contraction actually refers to the multiplicative factor that describes how fast the distance to optimum decays. E.g. for 1d quadratics, gradient descent gives x_{t + 1} - x* = (1 - \alpha h(x_t) ) (x_{t} - x*) with |1 - \alpha h(x_t) | as the contraction. In the appendix of our upcoming new manuscript, this concept will be helpful in demonstrating examples on the motivation of generalized curvature.
>
>
> Q: Lemma 2 seems to use the spectral radius of the momentum operator as the *robustness*. But how can it describe the robustness? More details are needed to understand this.
>
> *Robustness* of momentum operator means that momentum GD can achieve asymptotic linear convergence rate, which is *robust* to 1) the variation of generalized curvature of the landscape. 2) a range of different learning rate. Specifically, from Equ (4), we see constant spectral radius \sqrt{\mu} of operator A_t can imply asymptotic linear convergence rate \sqrt{\mu}. Lemma 2 gives the *condition* to achieve this rate. As discussed in the two paragraphs following Lemma 2, given momentum \mu is properly set based on the dynamic range of generalized curvature, the *condition* can be robustly satisfied 1) regardless of the variation of generalized curvature; 2) with a range of different value for learning rate.

---

### Official Review · AnonReviewer3 · 2017-11-27
**Misleading and shaky theoretical motivation/approach**

**Rating:** 4
**Confidence:** 5

**Review:**

The paper explores momentum SGD and an adaptive version of momentum SGD which the authors name YF (Yellow Fin). They compare YF to hand tuned momentumSGD and to Adam in several deep learning applications.


I found the first part which discusses the theoretical motivation behind YF to be very confusing and misleading:
Based on the analysis of 1-dimensional problems, the authors design a framework and an algorithm that  supposedly ensures accelerated convergence. There are two major problems with this approach:

-First: Exploring 1-dim functions is indeed a nice way to get some intuition. Yet,  algorithms that work in the 1-dim case do not trivially generalize to high dimensions, and such reasoning might lead to very bad solutions.

-Second: Accelerated GD does not benefit over GD in the 1-dim case. And therefore, this is not an appropriate setting to explore acceleration.
Concretely, the definition of the generalized condition number $\nu$, and relating it to the standard definition of the condition number $\kappa$, is very misleading. This is since $\kappa =1$ for 1-dim problems, and therefore accelerated GD does not have any benefits over non accelerated GD in this case.
However, $\nu$ might be much larger than 1 even in the 1-dim case.


Regarding the algorithm itself: there are too many hyper-parameters (which depend on each other) that are tuned (per-dimension).
And as I have mentioned, the design of the algorithm is inspired by the analysis of 1-dim quadratic functions.
Thus, it is very hard for me to believe that this algorithm works in practice unless very careful fine tuning is employed.
The authors mention that their experiments were done without tuning or with very little tuning, which is very mysterious for me.

In contrast to the theoretical part, the experiments seems very encouraging. Showing YF to perform very well on several deep learning tasks without (or with very little) tuning. Again, this seems a bit magical or even too good to be truth. I suggest the authors to perform a experiment with say a qaudratic high dimensional function, which is not aligned with the axes in order to illustrate how their method behaves and try to give intuition.

---

> ### Public Comment · (anonymous) · 2017-12-20
> **Part III: A single global learning rate and momentum; demonstration on high dimensional quadratics**
>
> Q: Regarding the algorithm itself: there are too many hyper-parameters (which depend on each other) that are tuned (per-dimension).
>
> As mentioned in the abstract, YellowFin only auto-tunes two hyperparameters for the entire model: a single global momentum and a single global learning rate (i.e. YellowFin does not use per-dimension learning rates and per-dimension momentums). The main question in the intro, which motivates our paper, is “can we produce an adaptive optimizer that does not depend on per-variable learning rate adaptation?”. The Yellowfin tuning rule uses only one SingleStep instance in section 3.1 for the whole model (instead of using one instance for each variable). It operates on the high dimensional local quadratic approximation and uses estimates of extremal curvatures h_min and h_max over all possible directions. It solves for a single momentum and a single learning rate. Note the SingleStep problem is a direct generalization of our 1D analysis at the beginning of Section 3, by decomposing along the eigendirections of the high dimensional quadratic.
>
> We will make Section 3 more precise in an upcoming revision and emphasize that we run a single instance of the SingleStep optimizer for the entire model with the estimators for h_min and h_max providing rough estimates of extremal curvatures along all directions.
>
>
> Q: And as I have mentioned, the design of the algorithm is inspired by the analysis of 1-dim quadratic functions. Thus, it is very hard for me to believe that this algorithm works in practice unless very careful fine tuning is employed. The authors mention that their experiments were done without tuning or with very little tuning, which is very mysterious for me.
>  In contrast to the theoretical part, the experiments seems very encouraging. Showing YF to perform very well on several deep learning tasks without (or with very little) tuning. Again, this seems a bit magical or even too good to be truth. I suggest the authors to perform a experiment with say a quadratic high dimensional function, which is not aligned with the axes in order to illustrate how their method behaves and try to give intuition.
>
> Following up on AnonReviewer3’s suggestion, we demonstrate the convergence behavior on a high-dimensional quadratic which is not aligned with the axes. More specifically, we first generated a 1000d quadratic, with curvature 1, 2, 3, …, 1000 on the axes. Then we rotate the quadratic for 45 degrees on the planes defined by the axes with curvature i and 1001 - i (for i = 1, 2,3, …, 500) in a pairwise fashion. We assume SingleStep problem has access to the oracle to exactly measure the curvature range and the distance to optimum. As shown in Figure (https://github.com/AnonRepository/YellowFin_Pytorch/blob/master/plots/1000d_quadratics.pdf), YellowFin’s tuning rule can demonstrate linear convergence rate on this landscape using the full gradient.
>
> We have also set up anonymized YellowFin repositories with PyTorch and TensorFlow implementations. Our experiments can be easily replicated with the code in the repository.
> [PyTorch repo] https://github.com/AnonRepository/YellowFin_Pytorch
> [Tensorflow repo] https://github.com/AnonRepository/YellowFin
>
>
> In summary, we
>
> -- showed how 1D analysis can generalize to multidimensional cases on quadratics.
>
> -- showed that there exists a meaningful definition of a generalized condition number (GCN) for 1D functions; and gave simple 1D examples where acceleration is possible.
>
> -- provided requested demonstration on high dimensional quadratics, as well as anonymized code repositories to replicate results in our manuscript.
>
> We appreciate AnonReviewer3’s time and detailed comments/suggestions to further clarify our contributions, and merit of our optimizer. That said, we are happy to provide further analysis, clarifications and experiments to resolve further questions.

---

> ### Public Comment · (anonymous) · 2017-12-20
> **Part II: Generalized condition number and examples of acceleration on 1D**
>
> Q: Second, accelerated GD does not benefit over GD in the 1-dim case. And therefore, this is not an appropriate setting to explore acceleration. Concretely, the definition of the generalized condition number $\nu$, and relating it to the standard definition of the condition number $\kappa$, is very misleading. This is since $\kappa =1$ for 1-dim problems, and therefore accelerated GD does not have any benefits over non accelerated GD in this case. However, $\nu$ might be much larger than 1 even in the 1-dim case.
>
> In this response, we give a detailed explanation and examples about why:
> (i) our definition of a generalized condition number (GCN) is meaningful on 1D functions and;
> (ii) even on 1D, we can use the GCN to tune non-zero values of momentum and achieve faster convergence that with 0 momentum.
>
> Regarding the motivation of generalized curvature, our main idea is that the condition number, which is defined based on a local definition of curvature, can be generalized to incorporate longer-range, non-local variations of curvature. Specifically, classic curvature at a given point is described by the eigenvalues of the local Hessian. On quadratic problems, this curvature plays a crucial role on the rate of convergence: if we use learning rate α, and no momentum, on a 1D quadratic of curvature h, the contraction (i.e. the multiplicative factor describing the stepwise shrinkage of the distance towards optimum) at every step is |1-αh|. The intuition is that when h is high, the gradient output exerts a strong ‘pull’ towards the optimum. Unfortunately, when we move to non-quadratic problems, this tight connection between curvature and convergence rate is lost.
>
> Our definition of ‘generalized curvature’ tackles this issue and maintains this tight connection for non-quadratic problems. We define it with respect to a specific local minimum and it describes how strong the ‘pull’ towards that minimum is. If h’(x) describes the generalized curvature at point x, the contraction for gradient descent with no momentum becomes |1-αh’(x)|. In this we regain the tight connection between our new definition of curvature and the convergence rate.
>
> As the first simple example to show acceleration in 1D, let's consider the 1D function f(x)=|x|. Curvature for all x \neq 0 is 0. Generalized curvature on the other hand is h’(x) = 1/|x|. Now if we restrict ourselves to x \in [ε, 1], the generalized (i.e. long-range) condition number (GCN) (with optimum x* = 0) is 1/ε. That is, as we get closer to the optimum, the relative pull towards it grows and the contraction factor becomes |1-α/|x||. Assume we are aiming for an accuracy of ε, then in the absence of momentum, we need to set the learning rate as α=Ο(ε). This means that starting from x_0=1, our first steps are going to converge at a rate of ~|1-ε|, which can be extremely slow. On the other hand, if we use the GCN to tune our momentum and learning rate, we get a momentum μ = 1-2(sqrt(ε)/(1+sqrt(ε)) ~= 1-2sqrt(ε), and experience a constant rate of sqrt(μ)=sqrt(1-2sqrt(ε)). As shown in the following plot (https://github.com/AnonRepository/YellowFin_Pytorch/blob/master/plots/contraction_abs_func.pdf), momentum is able to achieve stronger contraction, i.e. faster reduction of the distance to the optimum, during the first steps of optimization.
>
> Furthermore, the next figure (https://github.com/AnonRepository/YellowFin_Pytorch/blob/master/plots/convergence_abs_func.pdf) uses the analytical results above over time and shows that for f(x)=|x| using momentum tuned according to our GCN, can yield acceleration over gradient descent without momentum on 1D functions.
>
> As a second example to show acceleration in 1D case, we would like to point to the non-convex example of Figure 3(a) of our manuscript. In that case again, GCN=1000 which suggest a value of momentum of about 0.9, even though this is a 1D function. Plot 3(b) already shows that using this momentum allows for a constant linear convergence rate on this non-convex function. Assume in Figure 3(a), that the curvature of the top, flatter quadratic is 1 and the curvature of the bottom, steeper quadratic is 1000. The learning rate for gradient descent without momentum cannot exceed 1/500, otherwise it would always escape from the steep quadratic. Again, a similar analysis to the one we presented in the previous examples, as well an simple experiments, show that the optimal value for momentum again is not zero, even though this is a 1D function.
>
> The reason we are able to achieve this acceleration, is because we are taking into account long-range variations in curvature.

---

> ### Public Comment · (anonymous) · 2017-12-20
> **Part I: Scope of our work, and generalizing from 1D to multidimensional analysis**
>
> We appreciate AnonReviewer3’s helpful and detailed comments. *We will upload a new manuscript with the clarification incorporated in the next couple of days*. In the following, we address AnonReviewer3’s questions in detail. In summary, we:
>
> -- elaborate that the goal of the present paper is not to provide theoretical guarantees for general convex functions, but to use our simple analysis from quadratics to design an optimizer that works well empirically.
>
> -- explain how 1D analysis can generalize to multidimensional analysis in our case.
>
> -- explain why there exists a meaningful definition of a generalized condition number (GCN) for 1D functions; we give simple examples where accelerated linear rate is possible for 1D cases when we use the GCN (instead of conventional condition number) to tune momentum.
>
> -- provide the requested demonstration of convergence behaviors on high dimensional quadratics, as well as anonymous repo for experiment replication.
>
>
> Q: I found the first part which discusses the theoretical motivation behind YF to be very confusing and misleading:
> Based on the analysis of 1-dimensional problems, the authors design a framework and an algorithm that  supposedly ensures accelerated convergence.
>
> We would like to clarify that our algorithm does not ensure convergence outside the class of quadratic functions for multiple dimensional cases. This difficulty to guarantee general results for (Polyak’s) momentum gradient descent has been documented by the existence of specific counter-examples (Lessard et. al, 2016). We cite this paper in section 2 in order to make it clear that we do not give general guarantees.
>
> Instead we focus on ideas inspired from quadratics; these drive the design of our tuner, our main contribution. We conducted extensive experiments on 8 different popular deep learning models, empirically demonstrating our tuner’s merit on non-quadratics. We will make this point more prominent in the manuscript.
>
>
> Q: There are two major problems with this approach: First: Exploring 1-dim functions is indeed a nice way to get some intuition. Yet,  algorithms that work in the 1-dim case do not trivially generalize to high dimensions, and such reasoning might lead to very bad solutions.
>
> We agree with the reviewer that this is not an obvious point and elaborate here. We are updating our manuscript with the following clarifying discussion. We start by discussing a blueprint for generalization that is exact for quadratics (by extending discussions in the paragraph above Lemma 4). In our answer to the next question, we extend some of the quantities and ideas---like (generalized) curvature and (generalized) condition number---to the non-quadratic case. Then we follow the same generalization blueprint to go from 1D to multidimensional. As noted, we do not aim to provide analytical guarantees for non-quadratics, but rather to design an adaptive optimization method that works empirically well.
>
> The analysis of momentum dynamics on quadratic objectives decomposes exactly into independent scalar problems along the eigenvectors of the Hessian. For each scalar quadratic problem, the curvature is constant everywhere; we can think of this as the degenerate case where the extremal curvatures are equal, that is h_min=h_max. As the reviewer points out, the condition number for individual slices is 1, hence the optimal momentum value is 0.
>
> All scalar components of the multi-dimensional quadratic can be tuned jointly using a single instance of SingleStep to yield a single learning rate and single momentum value. In that case extremal curvatures h_min and h_max are taken over all directions and their ratio yields the condition number; individual 1D gradient variances sum up to the total multi-dimensional gradient variance, C, and the squared distance from optimum, D^2, can be either calculated as the sum of squared distances on the scalar problems, or approximately estimated directly (the latter is what we do in our implementation of the Distance() function). In this rebuttal we include a sanity-check experiment on a synthetic quadratic. It shows that when using exact oracles as input to SingleStep, we achieve the optimal convergence rate for quadratics.
>
> Now that we have established a blueprint for going from 1D to multidimensional analysis, in our next answer, we give a detailed explanation and examples about why:
> (i) our definition of a generalized condition number is meaningful on 1D functions and;
> (ii) even on 1D, we can use it to tune non-zero values of momentum and achieve faster convergence that with 0 momentum.

---

### Official Review · AnonReviewer2 · 2017-12-03
**Worthy contribution**

**Rating:** 6
**Confidence:** 1

**Review:**

[Apologies for short review, I got called in late. Marking my review as "educated guess" since i didn't have time for a detailed review]

The paper proposes an algorithm to tune the momentum and learning rate for SGD. While the algorithm does not have a theory for general non-quadratic functions, experimental validation is extensive, making it a worthy contribution in my opinion. I have personally tried the algorithm when the paper came out and can vouch for the empirical results presented here.

---

> ### Public Comment · (anonymous) · 2017-12-20
> **Thanks for supporting the validity of our experiments**
>
> We thank AnonReviewer2 for providing independent support on the accuracy of our experiments. The kind support on the validity of our experiments is a great encouragement to us.

---

### Public Comment · ~Carlos_Stein_Brito1 · 2018-01-05
**This method is a learning rate scheduler and it's unrelated with momentum**

I read the paper carefully and I don't think the results are related with momentum at all. In Figure 5 one sees that the momentum value does not go above 0.8, which shouldn't make much difference (common values are 0.99 for example).
I confirmed this by turning off momentum and rescaling the learning rate appropriately, getting the same results. Plot:
https://github.com/cstein06/YellowFin/blob/no_momentum/cifar/results/compare_losses.pdf

It means that it is a learning rate scheduler in disguise.

---

> ### Public Comment · (anonymous) · 2018-01-05
> **Adaptive momentum tuning is important in the YellowFin framework**
>
> Dear Carlos,
>
> thank you for the interest in our method! You pose a salient question and we have the experimental results to answer it in the appendix of our manuscript.
>
> It is absolutely true that positive momentum is *not always* necessary to achieve the best performance possible in the SGD+Momentum framework. Your plot is a good example of that (we’d like to point out that momentum tuning does not hurt performance in this case, if anything it marginally improves things).
>
> However, most importantly, *in many cases, adaptively tuning momentum strictly improves performance over the prescribed constant momentum values (such as 0.0 or 0.9).*
>
> Our manuscript includes experiments in Figure 10 (Appendix J.2) in support of this point. Specifically, we performed the following experiment: fix the momentum value to either 0.0 or 0.9 and just use the tuned learning rate from YellowFin. The results show an example where momentum tuning makes a big difference throughout training (the CharRNN on the left) and another example (the CIFAR100 Resnet on the right) where fixed 0.0 momentum seems optimal in the beginning, but eventually loses out to the adaptive momentum curve (the red YF curve).
>
> In summary, adaptively tuning momentum:
> does not seem to be hurting performance in your example (or in the problems we considered)
> strictly improves performance in many cases we’ve seen (and included in our manuscript)
>
> Best regards,
> The authors

---

> > ### Public Comment · ~Carlos_Stein_Brito1 · 2018-01-05
> > **Needs to rescale learning rate**
> >
> > Notice that I turn momentum off, but also rescale the learning rate in order to have the same effective learning rate. In this case the learning rate is not constant anymore, but there is no momentum. In my code, the important change is
> >
> > self._optimizer = tf.train.GradientDescentOptimizer(1.0 * self._lr_var * self.lr_factor / (1.0 - self._mu_var))
> >
> > in https://github.com/cstein06/YellowFin/blob/no_momentum/tuner_utils/yellowfin.py
> >
> > In this case there is no momentum, but the results should be very similar in general. The figure I posted is for the CIFAR + Resnet model in your paper.

---

> > > ### Author Response · Authors · 2018-01-12
> > > **YellowFin with non-zero momentum can demonstrate better and more stable performance**
> > >
> > > Dear Carlos,
> > >
> > > Thanks for the clarification. Rescaling the learning rate as per your suggestion, on multiple experiments, has the following consequences:
> > >
> > > -- As expected, training is sped-up compared to using YF’s LR un-adjusted and just setting momentum to 0. In some cases, like the one you conduct experiment on, YF is only marginally better (or the same) than YF-minus-momentum-plus-rescaling.
> > >
> > > -- Signs of instability start showing. Essentially, trading momentum for higher learning rate when we do the suggested rescaling, makes the performance curves behave more unstable (figures below).
> > >
> > >
> > > Overall, in a number of examples, the original, momentum based YF tuner demonstrates better validation metrics and better stability than the suggested rescaling rule with zero momentum.
> > >
> > > E.g. YellowFin rule with non-zero momentum can demonstrate better validation perplexity in the constituency parsing model. (https://github.com/AnonRepository/YellowFin_Pytorch/blob/master/plots/parsing_test_perp.pdf).
> > >
> > > In the following example of ResNext on CIFAR10, YellowFin rule with non-zero momentum can also demonstrate more stable validation accuracy (https://github.com/AnonRepository/YellowFin_Pytorch/blob/master/plots/cifar_smooth_test_acc.pdf).
> > >
> > > As the third example, using the momentum-based tuner, we are able to boost the learning rate (as described in Appendix J4) to get even better performance. In this example, we use learning rate factor 3.0 to increase the learning rate on ResNext for CIFAR10 (as what we did in the experiments in appendix J4), YellowFin rule with non-zero momentum gives *both* observably higher and more stable validation accuracy than the suggested rescaling rule (https://github.com/AnonRepository/YellowFin_Pytorch/blob/master/plots/cifar_smooth_test_acc_lr_fac_3.pdf ).
> > >
> > > We appreciate the feedback on this important point! We will be adding this discussion to our manuscript and, would also be happy to add an acknowledgement for your suggestion.
> > >
> > > Best regards,
> > > The authors

---

### Author Response · Authors · 2018-01-09
**Updated manuscript**

Dear readers and reviewers,

we have updated our manuscript during the rebuttal period in addition to our response to the official reviews. Specifically, we:

1. performed a significant rewrite of Sections 2 and 3 to make exposition of our ideas much more clear.

2. we added discussion in section 3.1 on how our tuning rule in equ(8) intuitively generalizes to multiple dimensions.

3. we addressed a number of reviewers comments and suggestions on clarification.

Best regards,
The authors

---

### Decision · Program_Chairs · 2018-01-29
**ICLR 2018 Conference Acceptance Decision**

**Decision:**

Reject

**Comment:**

This paper asks when SGD+M can beat adaptive methods such as Adam, and then suggests a variant of SGD+M with an adaptive controller for a single learning rate and momentum parameter.  There is are comparisons with some popular alternatives.  However, the bulk of the paper is concerned with a motivation that didn't convince any of the reviewers.